# Identifying Emissions Reduction Opportunities in International Bilateral Emissions Trading Systems to Achieve China’s Energy Sector NDCs

**DOI:** 10.3390/ijerph20021332

**Published:** 2023-01-11

**Authors:** Jing Han, Weilin Zhu, Chaofan Chen

**Affiliations:** School of Economics and Resource Management, Beijing Normal University, Beijing 100875, China

**Keywords:** Paris Agreement, emissions trading system (ETS), nationally determined contributions (NDCs), emissions reduction

## Abstract

Exploring more emissions reduction opportunities for China’s energy sector and lowering China’s decarbonisation costs are essential to fulfilling China’s nationally determined contributions (NDCs) and making China’s sustainable development more feasible. This study explored emissions reduction opportunities for China’s energy sector in international bilateral emissions trading systems (ETSs) using a CGE (computable general equilibrium) model. This study revealed that linking China’s ETS to those of regions with lower decarbonisation responsibilities, which tend to be developing regions, could lower China’s carbon prices, thus increasing China’s domestic energy supply and lowering energy prices (and vice versa). Meanwhile, the volume of emissions from regions linked to China also significantly affected the degree of the change in China’s carbon prices. Among these, ETS links to India and Russia could reduce China’s carbon price from 7.80 USD/ton under domestic ETS to 2.16 USD/ton and 6.79 USD/ton, allowing the energy sector and energy-intensive sectors to increase greenhouse gas emissions by 1.14% and 7.05%, respectively, without falling short of meeting its NDC targets. In contrast, as a consequence of links with the United States and the European Union, China’s carbon price could increase to 5.37 USD/ton and 1.79 USD/ton, respectively, which would limit China’s energy and energy-intensive sectors to emitting 5.45% and 2.24% fewer greenhouse gases in order to meet its NDC targets.

## 1. Introduction

The rapid transition towards clean energy in China triggered an energy crisis in September 2021, causing shortage in electricity supplies in many regions across China and prompting the government to slow its emissions reduction strategies [1]. However, China’s contributions to global emissions mitigation are critical since it is the world’s largest emitter of greenhouse gases; however, its economic development is also crucial. Global warming that has been driven by human-caused phenomena, such as greenhouse gas emissions and deforestation, has historically resulted in numerous significant issues [2]. Desertification, glacier retreat, resource depletion, species extinction and natural disasters, such as extreme weather events, have all recently threatened human life and water and food security [3,4,5,6]. The World Health Organization [7] noted that global warming is the greatest threat of the 21st century. To avoid catastrophic global climate change, greenhouse gas emissions must be reduced. In December 2015, several countries adopted an international climate agreement during the United Nations Framework Convention on Climate Change (UNFCCC) Conference in Paris. Since then, 194 parties have submitted their NDCs, which detail each country’s efforts to reduce its emissions by 2020 and every five years thereafter (i.e., 2020, 2025, and 2030).

Since the Paris Agreement was signed in 2015, many policies and laws have been enacted in China to encourage decarbonisation and the transition towards renewable energy sources. China is increasing its forest cover faster than any other country and it has become a leader in developing environmentally friendly technologies [8]. China produces more wind and solar power than any other country and it is the largest buyer of electric vehicles. China is also the world’s third largest consumer of natural gas [9]. The improvements in energy efficiency in China are primarily attributable to these policies, which have led to a decrease of 18.8% in the intensity of carbon emissions in China over the past five years [10]. While China has reduced its emissions intensity, some experts still claim that the government is not doing enough to combat climate change [11].

Furthermore, the economic development of China is of critical importance to the country and the rest of the world. As the world’s largest developing country, China has proven to be a significant stabilising force in the global economy’s recovery following the COVID-19 outbreak and its expanding market has presented numerous business opportunities [12]. However, China is faced with a myriad of serious challenges, including extricating itself from the middle-income trap and dealing with an ageing population, pollution, and trade barriers [13,14]. Meanwhile, some of China’s coal-fired power generators are relatively new and the early retirement of these investments prior to their final depreciation could result in substantial sunk costs for China, which is unacceptable to investors [15,16].

Therefore, it is necessary to find the most economical ways for China to mitigate its emissions, as well as finding more emissions reduction opportunities. Emissions trading has been proven to be an effective method for decarbonisation. For a given level of abatement, the US Environmental Protection Agency’s computer modelling has identified “cap and trade” systems as the solution with the lowest costs due to reductions in individual firm mitigation costs from each emitter deciding on the most economically efficient way to reduce emissions [17]. Emissions trading can reduce policy errors in the face of uncertainty, complementary policies, and international competition [18]. Nordhaus [19] noted that emissions trading systems could boost economic efficiency. Permit trading can also increase production and profit margins [20]. As a result, emissions trading systems can have significant impacts on green technology [21,22].

National or regional ETSs are widely used and have paved the way for international ETSs, which could offer China an opportunity to relax its intended NDC/INDC-imposed emissions limits. ETSs have long been discussed as offering a promising way to reduce emissions. In 2020, only 15.1% of global GHG emissions were covered by carbon pricing instruments; by 2021, this proportion had risen to 21.51% [23]. After years of testing, China’s national ETS went live in 2021. In recent years, there has been a wide range of discussions regarding the proposal to link ETSs between jurisdictions via the mutual or unilateral recognition of INDC emissions allowances, as outlined in Article 6 of the Paris Agreement [24]. In practice, an EU–Swiss ETS link was established in September 2020 and the Western Climate Initiative (WIC) carbon market now includes Virginia.

Utilising market-based instruments, such as ETSs, to help countries and regions to meet their NDC commitments is a growing area of research in the literature. However, studies examining proposals to link China’s ETS to the ETSs of other developing nations are lacking. Additionally, the economic mechanisms underpinning the effects of ETSs and ETS linking have not been thoroughly studied. The majority of studies have focused on linking China’s ETS with the ETSs of developed regions and have shown that these links could raise China’s energy prices and harm China’s GDP growth and welfare; therefore, these ETS connections may be unacceptable to China. This study determined which regions China should create ETS links with and analyses how the ETSs could affect China’s energy market and industries using the GTAP-E-PowerS model (a GTAP model that extensively covers both environmental and emissions factors, including electricity-level detail and the comprehensive inclusion of greenhouse gas emissions).

The following questions were explored in this study: Do international ETS (emissions trading system) links provide China’s energy sector with low-cost emissions reduction opportunities? Which regions could reduce emissions costs for China’s energy sector? Why would links to some regions reduce emissions costs for China’s energy sector while links to other regions would not? How could the changes in emissions costs resulting from ETSs and ETS links affect China’s domestic and international energy markets? Are the effects of ETS links sensitive to the development of green technology, such as the use of electricity to replace the use of fossil fuels?

## 2. Literature Review

The market-based mechanisms discussed in this study (namely, the cap and trade approach) cap emissions for some emitters while allowing them to trade emissions permits, thereby creating economic incentives for greenhouse gas emissions mitigation [25]. Governments either issue emissions permits based on baseline emissions or sell them at auctions. Emitters can then trade emissions credits to reduce the costs of meeting emissions targets [26]. Emitters with lower abatement costs are more likely to sell emissions credits and vice versa [27].

Linking ETSs creates larger carbon markets than those created by domestic ETSs, thereby increasing market liquidity and potentially lowering overall mitigation costs [28]. However, several studies have shown that linking existing ETSs does not always benefit all participants and can even cause welfare losses in the permit-selling region due to the effects of the terms of trade (TOT) [29,30,31]. Emissions credit importers tend to benefit from importing emissions permits, which eases production constraints, lowers carbon prices, and allows them to regain international competitiveness [32], whereas exporters of emissions permits can further reduce their own emissions and the production of sectors that generate large volumes of emissions [33]. According to simulations performed by Fujimori et al. [34], several regions, including China, were expected to suffer economic losses due to TOT effects.

However, because international ETS linking has not been widely established, there are limited observational data on its effects. Consequently, these effects cannot be evaluated using observational or econometric analyses; instead, simulation models, such as computable general equilibrium (CGE) models, are typically used to assess the effects of ETS links between jurisdictions [35,36,37,38,39]. Among these, the GTAP-E model has been widely used to evaluate the impacts of energy and climate policies [35,36,37,38,39,40].

While many studies have used CGE models to assess the effects of regional decarbonisation policies, only a few have examined emissions policies from outside the studied regions. A growing number of regions (and not just those considered in simulations in the literature) have adopted emissions mitigation policies. Due to the possibility of spillover effects from emissions policies outside the simulation area on the simulation results, ignoring those effects can yield inaccurate results. For instance, when simulating China’s decarbonisation policies, few studies have considered emissions mitigation policies from the rest of the world (e.g., those of the US or the UK) [31,41,42]. However, decarbonisation policies that have been implemented outside the simulation region can alter the terms of trade, thereby partially offsetting the negative leakage effects of policy enforcement within the simulation region. On the other hand, these policies can also reduce income, and, thus, market size outside the simulation region, thereby exacerbating the adverse effects. Furthermore, the mechanisms through which ETSs and ETS links act on economies, sector prices, and outputs have not been adequately explored.

This study attempted to fill the abovementioned research gaps and provide meaningful references for the adoption and study of carbon pricing, especially the linking of China’s ETS with those of other countries.

It is well acknowledged that a growing number of regions have plans to adopt emissions reduction policies, but previous simulations have tended to only consider the policies of a few regions and have assumed that most other regions have no emissions reduction policies, thus potentially overestimating the costs of China’s emissions reduction policies. Therefore, this study compared China’s losses from adopting an ETS in various scenarios, including considering the emissions targets of nearly all NDC submitters and only considering China’s NDCs, which produced results that were similar to those from a single-region CGE model.

This study also examined the ramifications of linking China’s ETS with those of other regions in order to determine which countries China should link with to reduce its energy consumption burden, thereby making China’s decarbonisation actions more feasible.

Since electrification is a trend within green development, this study also implemented a robustness test to examine the effects of recently developed electricity substitution technologies on the endogenous carbon prices in the ETS.

## 3. Model Framework and Simulation Design

### 3.1. Model

The model employed in this study was a GTAP-E-PowerS, which was modified from the GTAP model by Nong [43,44]. The GTAP model is a CGE model based on the theory of microeconomies and macroeconomies that uses model computable tools to systematically simulate policy influences using millions of equations. This kind of model was developed from a vast economic system of interlocking markets and was first put forward by Walras [45]. In the model of this study, a total of 385,436 equations were calculated using GEMPACK (General Equilibrium Modelling PACKage) version 12.1.

For each agent (producers, end users, exporters, importers, investors, etc.), the model simulated their behaviour and income flow resulting from price fluctuations caused by climate change mitigation policies. The model of this study simulated the behaviour of the agents using various economic (utility and production) functions and their parameters. The magnitude of the parameters in the behaviour functions indicated the similarity between the commodities and inputs in the behaviour functions. Constant elasticities of substitution (CES) functions were used to simulate agent behaviour. When decarbonisation policies affected the prices paid by final users (i.e., representative households and governments), the end users adjusted their purchasing decisions to maximise utility while adhering to their budget constraints. Leontief functions were used to group commodities and inputs with different functionalities, which are used in fixed proportions to their outputs (or requirements).

As with the original GTAP model, the GTAP-E-PowerS was built and categorised on the basis of economic behaviour. Consumption decisions are made using a linear expenditure system (LES), which is subject to changes in income and prices. Households (i.e., private households and governments) that have income remaining after their expenditures (e.g., consumption and income taxes) are saved. Taxing economic behaviour across sectors (i.e., production, consumption, investment, imports, and exports) generates revenue for governments, which use the Cobb–Douglas function to make public procurement decisions that are subject to changes in service and commodity prices.

The topmost structure in Figure 1 shows that producers process endowment–energy composites and other intermediate inputs to produce outputs using technology, as expressed by the Leontief function (fixed portion of input). The endowment–energy composites comprise capital–energy composites, labour, natural resources, and land with CES functions, which allows the producers to substitute inputs to lower production costs when facing the changing prices of inputs. The capital–energy composites are grouped into capital and energy elements using CES functions. The energy composites are categorised into electric and non-electric energy. The use of non-electric energy, especially coal, emits a great deal of greenhouse gases, so those who use a lot of non-electric energy need to purchase emissions quotas as complementary goods. Therefore, the endogenous carbon prices in the emissions trading system have a complementary influence on the consumption of coal. The use of electricity emits almost no greenhouse gases; therefore, carbon prices can encourage producers to use electricity to substitute emission-intensive energy with relatively clean energy and thus, carbon prices influence energy demand through the substitution effect. However, some electricity production processes, such as coal-fired baseload electricity production, generate large volumes of greenhouse gases, meaning that changes in carbon prices influence the production costs and, therefore, energy prices. Since emissions trading can influence production costs and trading revenue by changing carbon prices, it can also influence gross income and output demand.

### 3.2. Database

The foundation for the GTAP-E-PowerS model is a combination of the GTAP-E and GTAP-Power 10A datasets [46,47]. Although the GTAP-E database contains information on non-CO_2_ emissions, which comprise a sizable portion of GHG emissions, this information has not been integrated into the base data. Thus, this study adopted the method of Nong (2020) and incorporated non-CO_2_ emissions into the GTAP-E base data [44]. Additionally, following the work of Peter (2016) [48], this study broke down electricity, which is represented as an aggregated sector in GTAP-E version 10, into transmission and distribution (“TnD”) and 11 generation technologies (nuclear, coal, gas baseload, gas peak load, oil baseload, oil peak load, hydro baseload, wind, solar, and hydro peak load), which allowed us to account for those different subsectors within the electricity sector (“ely”) as they can have vastly different carbon policy responses due to substitution effects. The aggregated sectors simulated in this study are in Table 1.

To assess the effects of adopting NDCs as targets for meeting Paris Agreement commitments by the target year (2030), this study followed Siriwardana and Nong’s methodology and updated the GTAP-E-PowerS database, which uses 2014 as its base year rather than 2030, as specified in NDCs [39]. Consequently, this study updated our database from 2014 to 2030 using CEPII macroeconomic forest data on GDP, capital, population, and investments, in addition to data from the Centre for Global Trade Analysis on skilled and unskilled labour [49,50]. The updated data growth rates are shown in Table 2.

In total, 20 regions were selected as potential partners for China’s ETS based on their trade volume with China, GDP per capita, and data availability. Their ETSs were then individually simulated. Since Siriwardana and Nong predicted that developed regions would incur more significant abatement expenses than other regions, this study selected regions with varying levels of development for simulations of this study [39]. The World Bank categorised economies into four categories by GDP (which is a reliable measure of economic development) in 2021: low income, low–middle income, high–middle income, and high income [51,52]. According to UN trade data, this study selected the regions with the highest, highest middle, and lowest middle incomes to link with China’s ETS [53]. However, this study was limited in selecting low-income regions to link with China’s ETS in simulations of this study by the availability of other data, which meant that this study could only include two such regions.

Thus, this study simulates China (CHN) linking its ETS with the following regions shown in Table 3: the United States of America (USA), the European Union (EU), Japan (JPN), Korea (KOR), Singapore (SGP), and Australia (AUS) from the high-income group; Brazil (BRA), Mexico (MEX), Malaysia (MYS), the Russian Federation (RUS), Thailand (THA), and South Africa (ZAF) from the high–middle-income group; India (IND), Indonesia (IDN), Ukraine (UKR), the Philippines (PHI), Vietnam (VNM), and Iran (IRN) from the low–middle-income group; and Madagascar (MDG) and Uganda (UGA) from the low-income group. Based on the available data, this study considered a 2030 emissions mitigation target of 92.98% for NDC submitters and treated other NDC submitters with 2030 targets greater than their BAU as a single region (RNDC) in simulations.

Reporting on progress towards NDC goals lacks a unified system and a standardised approach [54]. Targets for reducing emissions can be reported in three different ways, depending on which is the most relevant for a given country: a percentage reduction from a baseline period; an absolute level of emissions; a percentage reduction from an assumed “business as usual” (BAU) level of emissions [55]. The International Monetary Fund (IMF) developed a methodology for calculating these targets and presenting them in a uniform and standardised format to make it easier to compare between countries. For our model, this study based our simulation goals on the IMF-estimated unconditional NDC targets for land use, land-use change, and forestry (LULUCF). These goals were derived from the IMF forecasts of emissions reduction, expressed as a percentage below “business as usual” (BAU) levels (Table 4).

### 3.3. Scenarios for Simulation and Macro Closures

Although energy production, residential energy use, and agricultural practices are all significant contributors to greenhouse gas emissions, only the energy and energy-intensive sectors are currently covered by carbon price policies in all major regions of the world. Therefore, the emissions trading systems simulated in our study only limited the emissions from the energy and energy-intensive sectors in all scenarios to meet the Paris Agreement target. Additionally, emissions from household and government energy consumption were assumed to be exempt from ETS regulations. Following the work of Siriwardana and Nong (2021), once an ETS was established in simulations of this study, emissions quotas were allocated to each sector at the uniform carbon prices of the ETS to which the sector belonged and at the same rate of decarbonisation as the NDC target submitted by the region to which the sector belonged [39]. Due to this rule, the initial allocation for each Chinese industrial sector was 93.38% of its baseline emissions level at a uniform price to fulfil China’s commitment under the Paris Agreement to reduce its overall emissions by 12.62%. Every sector covered by the same ETS was able to trade emissions quotas with each other freely. Income from ETS trading went into the domestic income. In this scenario, employment was determined by an external variable while endogenous variables determined the real wage rate, capital stock, and the rate of returns on capital. The long-run closure in the simulations was performed using Adams’s (2005) method [56]. This study created 22 scenarios that were broadly divided into three groups to examine the results of integrating ETSs (see Appendix A). For the first two types of scenarios, this study ran simulations to see whether this study needed to include carbon mitigation measures from regions that were not the focus of our analysis. Emissions permits were not transferable across international borders in those scenarios. The third set of scenarios was simulated to determine the impacts of integrating China’s ETS with those of the selected partners. All of the simulation results were aggregated and analysed to extrapolate the effects of ETSs on China’s carbon prices and energy market and the effects of China’s decision to link its ETS with those of regions with varying income levels.

The first type of scenario only included the “chqo” scenario, in which the domestic emissions trading system (ETS) was only implemented in China, i.e., the primary focus of our research, while the other regions were assumed to be free from decarbonisation policies. However, in practice, carbon policies in non-primary focus regions can impact another region’s carbon policies due to spatial spillover and carbon leakage effects [57,58].

The second type of scenario only included the “wrqo” scenario, in which the vast majority of NDC submitters with emissions targets lower than their baseline levels implemented domestic ETSs. With 92.98% of the 2030 baseline emissions of NDC submitters covered by the “wrqo” scenario, the emissions levels under their emissions reduction policies were significantly higher than in previous ETS studies. To see whether the simulation results for the region that was the primary focus of our research significantly differed from those that took into account the expansion of the scope of GHG emissions constraints, which could be more representative of the real future world, this study compared the “chqo” scenario to the “wrqo” scenario.

The third type of scenario built on the “wrqo” scenario by establishing bilateral international emissions trading systems (ETSs) and linking China’s domestic ETS with potential partners with various income levels. To create these scenarios, China’s ETS was linked with that of a partner country, creating a bilateral international ETS that allowed carbon permit trading across sectors. In the bilateral international ETSs, which were based on bilateral NDC agreements, this trading led to single permit prices. Countries with NDCs whose decarbonisation objectives were lower than their baseline targets were aware that the carbon prices in the bilateral international ETSs also applied to their energy and energy-intensive sectors.

## 4. Analysis and Discussion

The establishment of bilateral ETSs could change the decarbonisation responsibilities shouldered by China and its partner regions, thus further changing carbon prices in China and its partner regions. Changes in carbon prices can change domestic and international energy markets through complementary, substitution, cost, and income effects. Carbon prices can also change regional gross domestic production by changing the energy sector’s production costs.

### 4.1. Influence of the Implementation and Linking of ETSs on Carbon Prices

Linking bilateral ETSs could significantly alter regional decarbonisation responsibilities. Changes in decarbonisation responsibilities are also heavily influenced by the volume of emissions produced by regions linked to bilateral ETSs. If a region were linked to another region with relatively higher decarbonisation responsibilities, its own decarbonisation responsibilities would tend to increase, whereas if it were linked to another region with relatively lower decarbonisation responsibilities, its own decarbonisation responsibilities would tend to decrease.

As shown in Table 5, in simulations of this study, the percentage of NDC decarbonisation duties in China’s ETS increased to a relatively high degree when China established a bilateral ETS with the European Union or the United States but decreased to a relatively low degree when China established a bilateral ETS with Russia or India. The percentage of NDC decarbonisation duties of the ETS to which China belonged increased from 12.62% to 18.83%, 14.71%, 13.78%, 13.12%, 12.84%, 12.68%, 12.73%, 12.87%, 12.74%, and 13.45% when China linked its ETS with that of the United States, the European Union, Japan, Korea, Australia, Singapore, Brazil, Mexico, Thailand, and Indonesia, respectively. The percentage of NDC decarbonisation duties of the ETS to which China belonged decreased to 11.40%, 10.55%, 11.90%, 2.91%, 12.53%, 12.56%, 12.36%, and 12.36% when China linked its ETS with that of Malaysia, Russia, South Africa, India, the Philippines, Vietnam, the Ukraine, and Iran, respectively. It is evident that the decarbonization responsibilities shifted to a greater degree when China’s ETS was linked with countries with high emissions volumes. Even though Japan had more stringent decarbonisation requirements than the European Union, the total volume for trading within Japan’s domestic ETS was lower, so China’s decarbonisation requirements increased by a smaller amount when it linked with Japan’s domestic ETS.

The changes in the endogenous carbon prices of the selected markets, as a result of alterations to the decarbonisation duties brought about by establishing new carbon trading markets through the linking of various ETSs, are depicted in Figure 2. In regions with higher percentages of decarbonisation duties, the endogenous domestic carbon prices are typically higher when a domestic ETS was implemented. In this scenario, carbon prices in China, the United States, the European Union, Japan, Korea, Australia, Singapore, Brazil, Mexico, Thailand, Indonesia, the Philippines, Vietnam, Iran, Uganda, and Madagascar were 7.80 USD/ton, 60.39 USD/ton, 81.97 USD/ton, 190.31 USD/ton, 44.29 USD/ton, 43.64 USD/ton, 215.41 USD/ton, 67.49 USD/ton, 40.51 USD/ton, 12.64 USD/ton, 64.15 USD/ton, 3.17 USD/ton, 7.21 USD/ton, 6.50 USD/ton, 0.78 USD/ton, and 26.63 USD/ton, respectively. Regarding the regions whose IMF-estimated BAU greenhouse gas emissions levels did not exceed their emissions targets, such as Malaysia, Russia, South Africa, India, and Uganda, there was no carbon trading price. Nevertheless, this did not exclude the possibility that they have implemented carbon pricing policies that could have significant impacts on the IMF’s baseline emissions estimates for these regions; for example, the South African government has enacted a carbon tax. This study was similar to others that have looked into creating unified carbon markets between China and developed nations and have found that doing so would cause China’s carbon prices to increase [41,44,59]. However, previous studies have not explored the issue of establishing carbon markets between China and developing countries. The simulations in this study showed that building bilateral carbon markets with developing countries would likely reduce China’s carbon prices.

However, simulations of this study also showed that carbon permit prices decreased when an ETS was linked to another ETS with relatively lower carbon prices and increased when an ETS is linked to another ETS with relatively higher carbon prices. Therefore, linking China’s ETS to those of more developed regions increased the costs of carbon permits for China as more developed regions had greater decarbonisation duties. The trading volume of the linked regions impacted the endogenous carbon prices by influencing the percentage of decarbonisation duties. China faced a carbon price of 7.80 USD/ton when it implemented a domestic ETS. However, when it linked its ETS with those of the United States, the Europe Union, Japan, Korea, Australia, Singapore, Brazil, Mexico, Thailand, and Indonesia, the carbon prices increased to 13.29 USD/ton, 9.76 USD/ton, 8.88 USD/ton, 8.26 USD/ton, 8.08 USD/ton, 7.83 USD/ton, 7.92 USD/ton, 8.11 USD/ton, 7.86 USD/ton, and 8.55 USD/ton, respectively. In contrast, when China links its ETS to those of Malaysia, Russia, South Africa, India, the Philippines, Vietnam, Ukraine, Iran, and Uganda, its carbon prices decreased to 7.02 USD/ton, 6.79 USD/ton, 7.37 USD/ton, 2.16 USD/ton, 7.77 USD/ton, 7.80 USD/ton, 7.64 USD/ton, 7.78 USD/ton, and 7.80 USD/ton, respectively. In addition, linking ETSs had significant impacts on the carbon prices the parent regions had to pay. When their ETSs were linked to China’s, the United States, the European Union, Japan, Korea, Australia, Singapore, Brazil, Mexico, Thailand, and Indonesia saw a decrease in their carbon price of 47.10 USD/ton, 72.21 USD/ton, 181.43 USD/ton, 36.03 USD/ton, 35.57 USD/ton, 207.58 USD/ton, 59.57 USD/ton, 32.40 USD/ton, 4.77 USD/ton, and 55.59 USD/ton, respectively. Conversely, by linking their ETSs with China’s, the carbon prices in Malaysia, Russia, South Africa, India, the Philippines, Vietnam, Ukraine, Iran, and Uganda increased by 7.02 USD/ton, 6.79 USD/ton, 7.37 USD/ton, 2.16 USD/ton, 4.59 USD/ton, 0.58 USD/ton, 7.64 USD/ton, 1.28 USD/ton, and 7.02 USD/ton, respectively.

It can be seen in Figure 3 that when China’s ETS was linked to a region with higher carbon prices, China decarbonised more and sold more carbon permits, while when it was linked to a region with lower carbon prices, China tended to buy carbon permits and emit more greenhouse gases. However, the implementation of ETSs and the linking of ETSs had very different impacts on the market roles of different sectors within different ETSs.

As shown in Figure 4, China’s emission-intensive industries, such as coal, coal baseload electricity, and gas peak load electricity, tended to decarbonise and become carbon emissions permit sellers when the country implemented an emissions trading system. However, this trend was reversed when China established bilateral ETSs with regions with lower carbon prices or fewer decarbonisation requirements, making carbon permits cheaper for China’s sectors. The degree to which carbon prices decreased then increased the likelihood of this happening. China’s coal-based electricity baseload industry was the largest buyer in the bilateral ETS between China and India, the second largest buyer in the bilateral ETSs between China and Malaysia and Russia, and the third largest buyer in the bilateral ETSs between China and Malaysia and Russia. However, when China established bilateral ETSs with Malaysia and India, the peak load gas electricity sector became a buyer and when China set up a bilateral ETS with India, all industries under the ETS participated as buyers.

Industries with lower greenhouse gas emissions (i.e., oil, gas, oil products, electricity transfer and distribution, non-fossil baseload electricity, oil peak load electricity, and energy-intensive secondary industries) became net buyers when China implemented an emissions trading system. When China’s carbon prices increased significantly because it linked its ETS with the ETS of a region with relatively higher carbon prices, this trend reversed. In our simulations, when China linked its ETS to those of the United States and the European Union, the energy-intensive secondary industries became significant sellers of carbon emissions permits. Establishing ETSs with China similarly transformed industry roles in partnership regions within the carbon market. Sectors in regions where carbon priced were lower than China’s under the domestic ETS scenario were more likely to become carbon permit sellers, while sectors in regions where carbon prices were higher than China’s were more likely to become carbon permit buyers when a bilateral ETS was established between the two regions. All sectors in the United States, Europe, Japan, Korea, Australia, Singapore, Brazil, and Mexico became buyers when bilateral ETSs were established with China. In contrast, all sectors in Malaysia, Russia, South Africa, India, and Uganda became sellers.

### 4.2. China’s Domestic Energy Market

As mentioned above, the endogenous carbon prices in China’s ETS changed by linking China’s ETS to those of other regions, which also influenced energy demand and energy prices through a variety of mechanisms, including complementary, substitution, cost, and income effects.

As shown in Figure 5, when only China implemented decarbonisation policy and other regions were free from decarbonisation policies, as in the “siqo” scenario, rather than considering most regions’ plans to decarbonise, as in the “wrqo” scenario, simulations of this study tended to overestimate the positive effect of carbon prices on energy demand and underestimate the negative impact of carbon prices on energy demand.

Figure 5 also illustrates whether and how linking China’s ETS to that of another region caused a decline in the demand for coal, coal baseload electricity, gas peak load electricity, oil peak load electricity, hydro peak load electricity, solar peak load electricity, and oil and an increase in the demand for nuclear baseload electricity, wind baseload electricity, hydro baseload electricity, and other electricity generated using clean energy sources. The demand for baseload electricity produced by clean energy sources, such as hydro baseload electricity, nuclear baseload electricity, and wind baseload electricity, increased the most out of all energy demand. In contrast, the demand for coal declined to the greatest extent and coal baseload electricity experienced the most significant decline. Additionally, oil demand dropped to the lowest degree.

The reduction in carbon prices caused by linking ETSs decreased demand for China’s energy compared to the domestic ETS scenario (“wrqo”). As carbon prices increased due to linking ETSs, Chinese energy demand also increased compared to the domestic ETS scenario (“wrqo”). Among all the linked ETS scenarios simulated in this study, the demand for China’s energy fluctuated the most when China established a bilateral ETS with the United States and the demand for China’s energy fluctuated the least when China established a bilateral ETS with India.

Figure 5 also illustrates that when a domestic ETS was implemented in all sectors, as in the “wrqo” scenario, endogenous carbon prices in the domestic ETS decreased demand for Chinese coal, oil, gas, and coal baseload power, gas peak load power, oil peak load power, hydro peak load power and solar peak load power by 13.63%, 0.60%, 5.53%, 9.96%, 7.60%, 6.42%, 6.09%, and 6.11%, respectively, but increased demand for clean Chinese energy from wind baseload electricity, hydro baseload electricity, and nuclear baseload electricity by 14.42%, 13.25%, and 14.48%, respectively. When China established a bilateral ETS with the United States, the increase in carbon prices had more significant complementary, substitution, cost, and income effects. In this scenario, the demand for Chinese coal, oil, gas, and coal baseload power, gas peak load power, oil peak load power, hydro peak load power, and solar peak load power decreased by 21.29%, 0.90%, 8.32%, 16.26%, 12.50%, 10.80%, 10.11%, and 10.13%, respectively, which was much more than in the “wrqo” scenario; meanwhile, the demand for Chinese wind baseload electricity, hydro baseload electricity, and nuclear baseload electricity increased by 22.51%, 20.50%, and 22.70%, which was also much more significant than in the “wrqo” scenario. In contrast, when China established a bilateral ETS with India, the decrease in carbon prices had lower complementary, substitution, cost, and income effects. In this scenario, the demand for Chinese coal, oil, and gas baseload power, gas peak load power, oil peak load power, hydro peak load power, and solar peak load power decreased by 3.59%, 0.29%, 1.50%, 1.45%, 2.47%, 1.77%, 1.26%, 1.33%, and 1.35%, respectively, which was much less than in the “wrqo” scenario; meanwhile, the demand for Chinese wind baseload electricity, hydro baseload electricity, and nuclear baseload electricity increased by 4.80%, 4.47%, and 4.74%, which was also much less than in the “wrqo” scenario.

There were two main mechanisms by which higher carbon prices increased energy prices: the income effect and the complementary effect. However, the exact mechanisms at play varied greatly depending on the context of the energy being considered. Increased carbon prices had a negative effect on domestic income and reduced gross demand for all energy via the income effect. In contrast, the complementary effect only lowered the prices of certain types of energy that emit a great deal of greenhouse gases but increased the usage costs due to the requirement to purchase emissions quotas, which were priced according to the increased carbon prices. As depicted in Figure 5, these two variables act in combination to reduce oil and gas prices in the Chinese market.

The substitution and cost effects could affect energy prices in distinct ways. The substitution effect of carbon prices tended to increase the prices of specific clean low-emission energy as their demand increased in response to increased carbon prices. Increases in the prices and demand for certain types of electricity, such as hydro baseload, wind baseload, and nuclear baseload power, reflected the substitution effect of carbon prices, as shown in Figure 5.

The cost effect of carbon prices tended to increase the prices of specific types of high-emission energy due to the requirement to purchase a more significant number of emissions quotas, which were priced according to the increased carbon prices, thereby increasing production costs. The increased prices of these types of energy further reduced demand for them. The cost effect of carbon prices was reflected in the increased prices of and decreased demand for coal-based baseload electricity, as shown in Figure 5.

These four carbon price effects caused the most significant fluctuations in the market prices of carbon-based baseload electricity in all scenarios and increased the costs of almost all energy sources, except oil and natural gas. When China’s ETS was linked to those of regions with higher carbon prices, energy price volatility increased. In contrast, energy price volatility decreased when China’s ETS was linked to those of regions with lower carbon prices.

The prices of coal baseload electricity, gas peak load electricity, coal, hydro baseload electricity, oil peak load electricity, wind baseload electricity, nuclear baseload electricity, solar peak load electricity, hydro peak load electricity, other clean baseload electricity, and electricity transfer and distribution increased by 20.64%, 4.91%, 3.80%, 2.23%, 2.07%, 1.48%, 1.44%, 1.34%, 1.30%, 1.26%, and 1.24%, respectively, in the “wrqo” scenario compared to the no-policy BAU scenario. In contrast, the prices of gas and oil decreased by 1.19% and 0.35%, respectively.

When China’s ETS was linked to that of the United States in the “chus” scenario, the increased carbon prices resulted in the most considerable increase in energy prices, including the price of coal baseload electricity, gas peak load electricity, coal, hydro baseload electricity, oil peak load electricity, wind baseload electricity, nuclear baseload electricity, solar peak load electricity, hydro peak load electricity, other clean baseload electricity, and electricity transfer and distribution, which increased 34.04%, 7.58%, 5.65%, 3.09%, 3.26%, 1.87%, 1.76%, 1.62%, 1.60%, 1.60%, and 1.41%, respectively. In contrast, the prices of gas and oil decreased by 1.38% and 0.75%, respectively.

When China’s ETS was linked to that of India in the “chin” scenario, the increased carbon prices resulted in an enormous increase in energy prices, including the prices of coal baseload electricity, gas peak load electricity, coal, hydro baseload electricity, oil peak load electricity, wind baseload electricity, nuclear baseload electricity, solar peak load electricity, hydro peak load electricity, other clean baseload electricity, and electricity transfer and distribution, which increased 6.30%, 1.92%, 1.60%, 1.15%, 0.74%, 0.93%, 0.97%, 0.90%, 0.86%, 0.80%, and 0.92%, respectively. In contrast, the prices of gas and oil decreased by 0.13% and 0.01%, respectively.

### 4.3. China’s International Energy Trade

Since the intensity and quantity of emissions from imported and exported energy were different, the costs of using these two kinds of energy were influenced by the endogenous carbon prices in the ETS to various degrees. The differences in costs resulted in the substitution of domestic and imported energy.

As shown in Figure 6, improper scenario design could result in inaccurate estimations about China’s energy trade. The “chqo” scenario, in which only China’s decarbonisation policies are assumed to be implemented, has been widely adopted by many simulations, especially single-region CGE models, and the prices of imported energy have been underestimated compared to those in other scenarios, such as “wrqo”, which accounts for the decarbonisation policies of nearly all regions. This conclusion differed from previous studies, which have predicted that the decarbonisation of more regions would limit carbon leakage through the international market effect [60,61]. The inaccurate estimation of imported energy prices could also result in the inaccurate estimation of the amount of energy imported. Import demand for primary energy sources (such as coal, oil, and gas) and secondary energy sources (such as baseload electricity generated by clean energy sources, including wind, hydro, and nuclear) were underestimated whereas import demand for secondary energy, such as coal-based baseload and peak load electricity, was typically overestimated.

The scenarios in which almost all regions fulfilled their NDCs by implementing ETSs had minimal effects on China’s importation of energy and the prices of imported energy, except for the “siqo” scenario. However, secondary energy imports, such as coal baseload electricity, gas peak load electricity, and oil peak load electricity, tended to become cheaper when China linked its ETS to those of regions with relatively higher carbon prices. In comparison, secondary energy imports became more expensive when China linked its ETS to those of regions with relatively lower carbon prices. The importation of emission-intensive energy and secondary energy, such as coal and coal-based electricity, tended to increase when China’s ETS was linked to those of regions with relatively high carbon prices and decreased when it was linked to those of regions with relatively low carbon prices. For example, when China established a bilateral ETS with the United States in the “chus” scenario, carbon prices significantly increased domestic energy prices, resulting in an increase in China’s coal baseload electricity importation from 5.88% in the “wrqo” scenario to 37.50% in the “chus” scenario.

### 4.4. Real GDP Changes from the Establishment of Linked ETSs

The implementation of ETSs and linking China’s ETS to those of other regions also changed China’s GDP by changing carbon prices and energy outputs.

As shown in Figure 7, the variations in the real GDP of the studied regions followed the same decarbonisation trends and the inverse carbon price trends. When China linked its ETS to those of regions with relatively higher carbon prices, China’s GDP decreased compared to in the “wrqo” scenario, while when China linked its ETS to those of regions with relatively lower carbon prices, China’s GDP increased. The number of emissions quotas and the amount of decarbonisation responsibility of a region affected the carbon price fluctuations in the regions with which it had established bilateral ETSs and, thus, affected GDP fluctuations in those regions.

When China set up bilateral ETSs with India and Russia, its GDP benefited the most. In contrast, when China set up bilateral ETSs with the United States and the European Union, its GDP suffered the most. China’s real GDP fell by 0.79% in the “wrqo” scenario, in which a domestic ETS was implemented. Real Chinese GDP also fell by 0.71%, 0.31%, 0.17%, 0.13%, 0.06%, 0.04%, 0.02%, and 0.01% when China linked its ETS with those of the United States, the European Union, Japan, India, Korea, Mexico, Brazil, Australia, Singapore, and Thailand, respectively. However, when China linked its ETS with those of India, Russia, Malaysia, South Africa, Ukraine, the Philippines, and Iran, China’s real GDP increased by 0.79%, 0.14%, 0.09%, 0.05%, 0.02%, 0.01%, 0.01%, 0.04%, 0.02%, and 0.01%, respectively, compared to that in the “wrqo” scenario. Moreover, establishing bilateral ETSs with Vietnam, Uganda, and Malaysia also significantly increased China’s real GDP.

### 4.5. Robustness Test: Electricity Substitute

Recent developments in China’s various kinds of electric technologies, such as electric cars, have made it easier for China to substitute electricity with non-electric energy when faced with fluctuations in different energy prices. Therefore, this study implemented a robustness test to verify whether our conclusions were sensitive to the development of green technology. The elasticity parameters of substitution within the CES function of a CGE model, which represent the ability to substitute between various commodities, are typically used by modellers to implement robustness tests of their conclusions [31,62]. Therefore, to test the robustness of the conclusions, this study altered the elasticity parameters of substitution between electricity and non-electric energy in the production function of the model by ±50% for all non-energy industries in China.

Firstly, this study tested the robustness of the endogenous carbon prices in China’s ETS, as shown in Figure 8. Even though more flexibility when substituting electricity with other types of energy resulted in slight decreases in the endogenous carbon prices in China’s ETS, the effects of linking China’s ETS to those of other regions on carbon prices was relatively robust. China’s carbon prices increased when China linked its ETS with those of regions with higher decarbonisation duties, whereas China’s carbon prices decreased when China linked its ETS with those of regions with relatively lower decarbonisation duties.

Secondly, this study tested the robustness of energy market fluctuations in China. The robustness tests showed that even though more flexibility when substituting electricity with other types of energy resulted in slight decreases in China’s energy prices and supply, linking China’s ETS to those of other regions had relatively robust impacts on carbon prices. When China’s ETS was linked with those of regions with higher decarbonisation duties, China’s energy prices and supply increased, whereas China’s energy prices and supply decreased when China linked its ETS with those of regions with relatively lower decarbonisation duties.

## 5. Conclusions

As the largest greenhouse gas emitter in the world, China’s decarbonisation is essential for the mitigation of global warming, which is the most serious problem facing the environment and global sustainable development. China has submitted nationally determined contributions to the United Nations, with the promise to decease its greenhouse gas emissions by 12.62% from its BAU scenario, as estimated by the IMF. This decarbonisation goal could greatly affect China, which has been working hard to recover from the economic damage caused by COVID-19 and also faces the challenges of an ageing population and the deterioration of international trading environments. Since September last year, some regions of China have suffered from shortages in electricity supplies, which were caused by decarbonisation actions. Furthermore, some of China’s coal-fired power generators are relatively new and the early retirement of these investments prior to their final depreciation could result in substantial sunk costs for China, which is unacceptable to investors.

Therefore, finding more emissions reduction opportunities for China’s energy sector, as well as keeping its NDC promise, could make China’s transition towards decarbonisation more feasible. Emissions trading systems (ETSs) have been demonstrated to be an efficiency market mechanism for decarbonisation. A domestic ETS covering the electricity sector was established in China last year. This study used a GTAP model that extensively covered both environmental and emissions factors, including electricity-level detail and the comprehensive inclusion of greenhouse gas emissions (GTAP-E-PowerS), to compare the effects of China implementing a unilateral ETS across China’s energy markets to the impacts of establishing bilateral ETSs between China and 20 other regions that have relatively good trade relationships with China.

The simulations demonstrated that international ETSs could provide China’s energy sector with the chance to emit more GHGs by buying low-cost emissions quotas from other regions and, thus, lower China’s carbon and energy prices, further lowering China’s decarbonization costs in terms of GDP. However, not all links to the ETSs of other regions provided this chance. As a matter of fact, China’s carbon and energy prices only decreased when China linked its ETS to those of regions with relatively lower decarbonisation duties, which also meant lower carbon prices. These regions tended to be lower income regions, while linking China’s ETS to those of regions with relatively higher carbon prices increased China’s carbon prices and, in turn, energy prices. Meanwhile, the volume of emissions from the regions linked with China also had a great effect on the degree of change in China’s carbon prices. For example, China’s carbon price may be reduced by ETS links with India and Russia from 7.80 USD/ton under domestic ETS to 2.16 USD/ton and 6.79 USD/ton, resulting in an increase of 1.14% and 7.05% in greenhouse gas emissions without falling short of meeting its NDC targets for the energy and energy-intensive sectors. However, China’s carbon price is likely to increase to 5.37 USD/ton and 1.79 USD/ton due to its links with the United States and the European Union. In order to meet its NDC targets, China’s energy and energy-intensive sectors would be required to emit 5.45% and 2.24% fewer greenhouse gases.

Even though the international emission trading system may provide a mechanism to low China’s decarbonization cost, much work still needs to be carried out to study China’s decarbonization. The idealized CGE model used in this study accurately reflects the trend of ETS linkage; however, there may be a slight discrepancy between the simulation result and reality due to the model’s assumptions concerning the perfect rationality, information, and foresight of actors (such as households and businesses) and the existence of perfect and complete markets. On the one hand, decarbonization is not only an economic problem but also a technology; as shown in the robustness test, the development of green technology can decrease the carbon price faced by China. On the other hand, improving the environment may also benefit China economically by easing the drought faced by some farmers beside the Yangtze River. Future studies need to include these benefits to evaluate the environmental benefits of emissions trading systems. If relevant data is available, future studies should also consider the transaction cost resulting from political negotiation. Although they are an expense with a relatively short time horizon, transaction fees play a crucial role in ETS’s efficient interconnection. This study looks into a wide variety of possible regions to ensure that if linkage with one region fails due to noneconomic reasons, including political reasons, other regions may still be able to link ETS with China.

## Figures and Tables

**Figure 1 ijerph-20-01332-f001:**
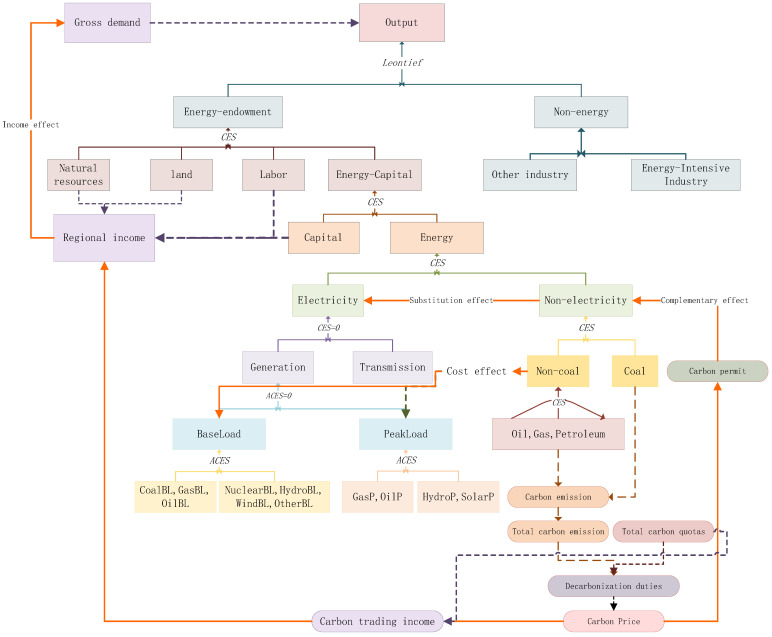
Structure and mechanisms underlying the effect of carbon prices on production.

**Figure 2 ijerph-20-01332-f002:**
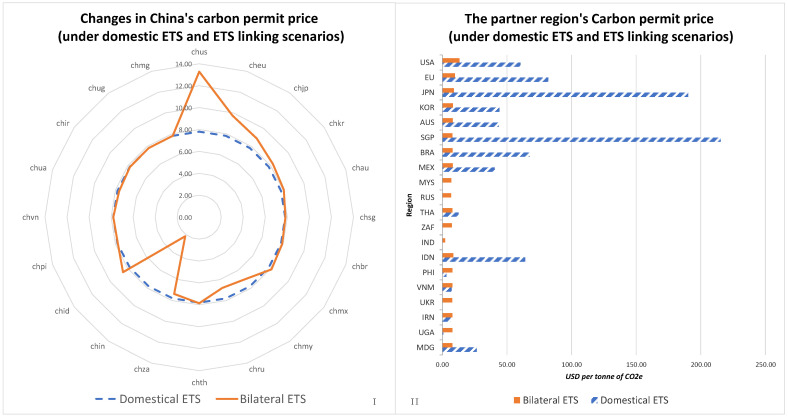
Fluctuating regional prices for carbon permits in various scenarios. (**I**,**II)**, under domestic ETS and ETS linking scenarios, carbon permit prices change in China (**I**) and the partner regions (**II**).

**Figure 3 ijerph-20-01332-f003:**
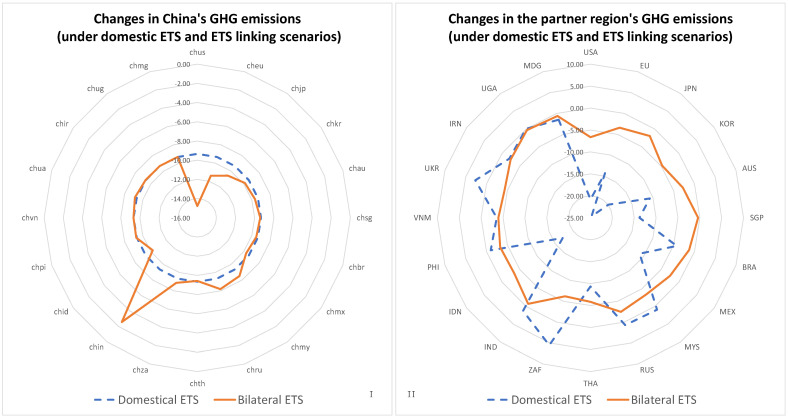
Changes in the greenhouse gas emissions of the different regions. (**I**,**II**), under domestic ETS and ETS linking scenarios, greenhouse gas (GHG) emissions change in China (**I**) and the partner regions (**II**).

**Figure 4 ijerph-20-01332-f004:**
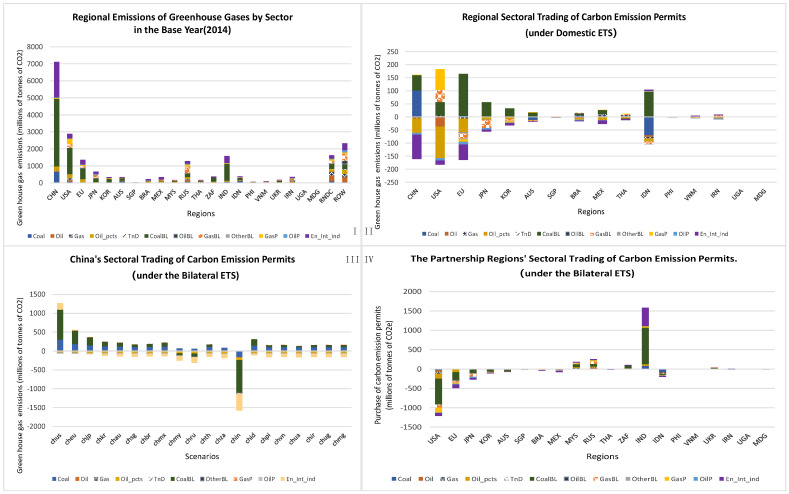
Emissions and carbon emissions permit trading at the regional level by sector. (**I**,**IV**), regional greenhouse gas emissions by sector in 2014 (**I**) and under the domestic ETS scenario in 2030 (**II**); and sectoral trading of carbon emission permits under the bilateral ETS scenario in China (**III**) and its partnership regions (**IV**).

**Figure 5 ijerph-20-01332-f005:**
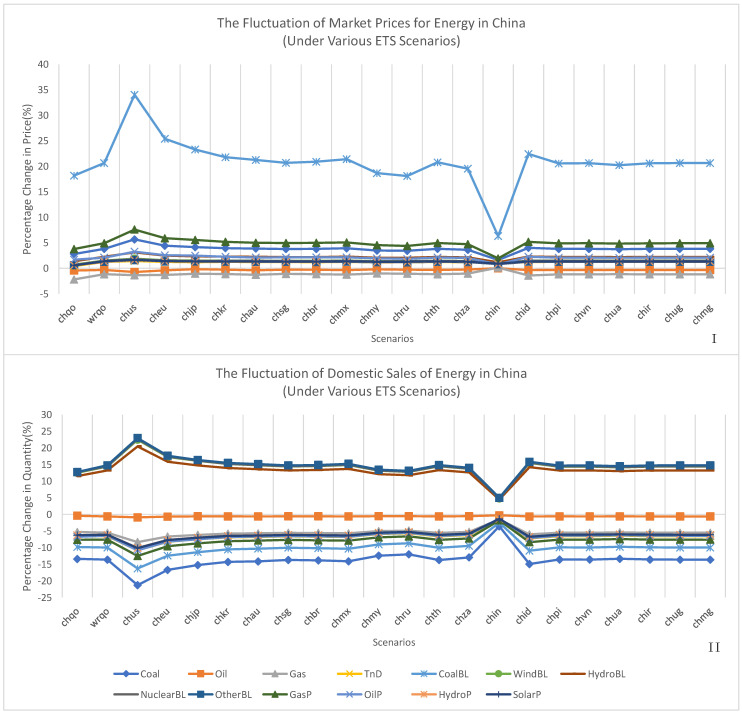
Fluctuations in domestic market prices and energy sales in China. (**I**,**II**), China’s market price fluctuation under domestic ETS and ETS linking scenarios (**I**), as well as China’s domestic energy sales under domestic ETS and ETS linking scenarios (**II**).

**Figure 6 ijerph-20-01332-f006:**
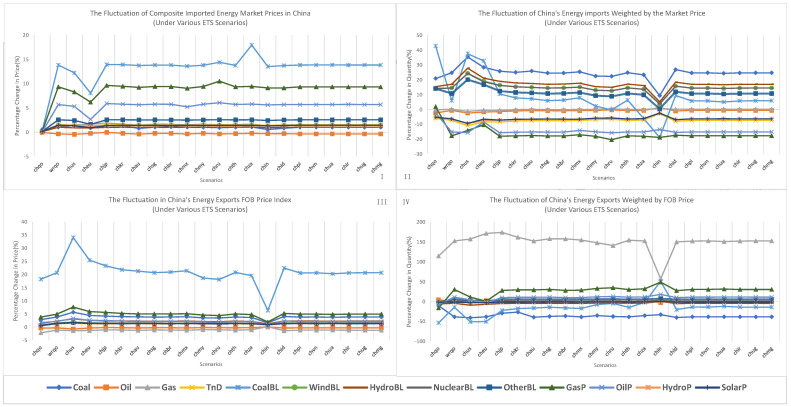
Fluctuations in China’s international energy trade. (**I**,**IV**), under domestic ETS and ETS linking scenarios, fluctuation of China’s imported energy market price (**I**), its energy imports weighted by the market price (**II**), its energy exports FOB price index (**III**), and its energy exports weighted by FOB price (**IV**).

**Figure 7 ijerph-20-01332-f007:**
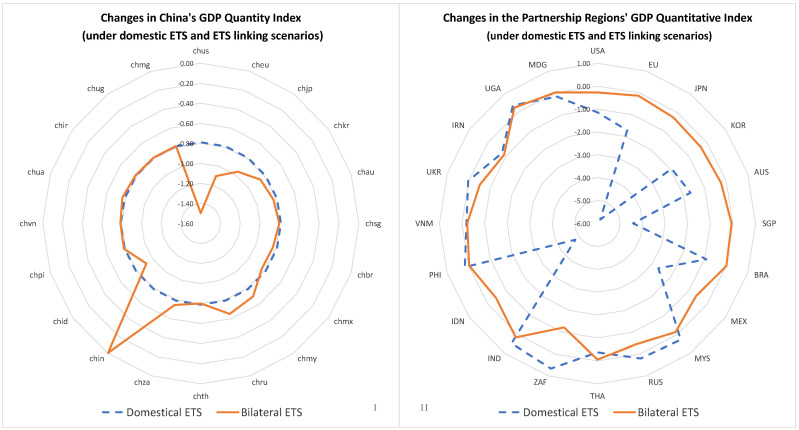
Changes in China’s real GDP in various scenarios of international bilateral ETSs. (**I**,**II**), Real GDP (gross domestic production) changes in China (**I**) and its partner regions (**II**) under domestic ETS and ETS linking scenarios.

**Figure 8 ijerph-20-01332-f008:**
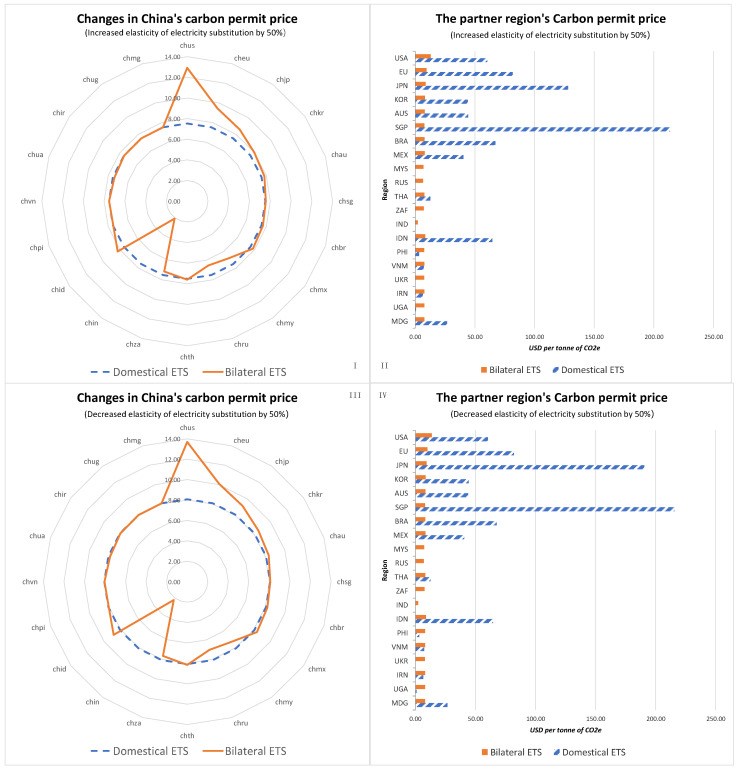
Robustness of the endogenous carbon prices in the ETS. (**I**,**IV**), carbon permit price change in China (**I**) and its partner regions (**II**) while the elasticity of electricity substitution increased by 50% under domestic ETS and ETS linking scenarios. Carbon permit price change in China (**III**) and its partner regions (**IV**) while the elasticity of electricity substitution decreased by 50% under domestic ETS and ETS linking scenarios.

**Table 1 ijerph-20-01332-t001:** Sectors used in the simulations.

Sector Number	New Sector Code	Comprising	New Sector Description
1	Agr	pdr wht gro v_f osd c_b pfb ocr ctl oap rmk wol frs fsh	Primary Agric., Forestry and Fish
2	Coal	coa	Coal Mining
3	Oil	oil	Crude oil
4	Gas	gas gdt	Natural gas extraction
5	Oil_pcts	P_c	Refined oil products
6	TnD	TnD	Electricity: Transmission and
7	NuclearBL	NuclearBL	Nuclear base load
8	CoalBL	CoalBL	Coal base load
9	GasBL	GasBL	Gas base load
10	WindBL	WindBL	Wind base load
11	HydroBL	HydroBL	Hydro base load
12	OilBL	OilBL	Oil base load
13	OtherBL	OtherBL	Other base load
14	GasP	GasP	Gas peak load
15	HydroP	HydroP	Hydro peak load
16	OilP	OilP	Oil peak load
17	SolarP	SolarP	Solar peak load
18	En_Int_ind	oxt chm bph rpp nmm i_s nfm	Energy-intensive industries
19	Oth_ind_set	cmt omt vol mil pcr sgr ofd b_t tex wap lea lum ppp fmp ele eeq ome mvh otn omf	Other industrial sectors
20	Services	wtr cns trd afs otp wtp atp whs cmn ofi ins rsa obs ros osg edu hht dwe	Services

Note: Except for agriculture (Agr), other industrial sectors (Oth_ind_ser), and services (Services), all other sectors in this study will be regulated by the carbon price system [39].

**Table 2 ijerph-20-01332-t002:** Projection of macroeconomic growth from 2014 to 2030 (percentage changes).

	GDP	Investment	Capital	Population	Unskilled Labour	Skilled Labour
CHN	135.6819	121.8243	205.9861	−2.86806	−4.734412	44.8284
USA	37.32951	35.69614	24.38688	14.02809	−1.533694	28.44798
EU	19.06453	17.02391	20.4287	4.69472	−22.49914	17.18812
JPN	19.43177	18.60576	18.00238	−4.132165	−20.42933	8.918434
KOR	32.62148	25.93604	21.567	−0.04060209	−15.44314	33.67079
AUS	52.72524	47.86806	29.25632	25.83834	−1.526394	27.61308
SGP	45.22276	40.55844	36.80211	9.174435	−34.32659	37.98201
BRA	6.233727	−2.113575	61.62977	5.858932	2.442562	50.79475
MEX	46.29286	42.94689	62.53793	8.690685	8.286116	54.02381
MYS	75.94946	71.18216	101.1069	19.60434	4.248103	85.25908
RUS	8.853173	4.875936	51.29598	−4.159203	−19.66497	3.3136
THA	61.43278	51.9866	105.4742	6.571364	−11.43758	48.98137
ZAF	11.07168	6.968047	52.71975	8.003175	−7.656701	57.33515
IND	185.519	176.7217	163.949	11.47663	19.37624	79.2283
IDN	99.87856	87.30717	121.9318	6.219156	11.22124	71.05772
PHI	147.7541	146.4606	108.4848	19.26004	27.6814	75.76202
VNM	123.8319	126.6005	98.8867	8.277095	9.784652	75.59243
UKR	−48.1177	−46.19702	30.81696	−9.319242	−16.81186	2.106369
IRN	68.70561	83.06553	108.8245	8.724825	−4.62567	46.07723
UGA	82.51842	67.75489	204.7832	47.60686	50.43645	127.8087
MDG	43.2833	31.2683	204.7832	30.32327	52.88404	130.7221
RNDC	48.9297	48.46354	40.46402	17.62543	27.89749	65.70395
ROW	59.01742	144.725	88.48922	21.38274	26.7676	79.05251

Note: calculate based on the data from CEPII and Center for Global Trade Analysis [49,50]; CHN, China; USA, the United States of America; EU, the European Union; JPN, Japan; KOR, South Korea; AUS, Australia; SGP, Singapore; BRA, Brazil; MEX, Mexico; MYS, Malaysia; RUS, Russia; THA, Thailand; ZAF, South Africa; IND, India; IDN, Indonesia; VNM, Vietnam; PHI, Philippines; UKR, Ukraine; IRN, Iran; UGA, Uganda; MDG, Madagascar; RNDC, the rest of the regions subjected to NDCs; ROW, the rest of the world.

**Table 3 ijerph-20-01332-t003:** Regions used in the simulations.

Sector Number	New Region Code	Comprising	New Region Description
1	CHN	chn	China
2	USA	usa	United States of America
3	EU	aut bel bgr hrv cyp cze dnk est fin fra deu grc hun irl ita lva ltu lux mlt nld pol prt rou svk svn esp swe	EU 27 member countries
4	JPN	jpn	Japan
5	KOR	kor	Korea
6	AUS	aus	Australia
7	SGP	sgp	Singapore
8	BRA	bra	Brazil
9	MEX	mex	Mexico
10	MYS	mys	Malaysia
11	RUS	ms	Russian Federation
12	THA	tha	Thailand
13	ZAF	zaf	South Africa
14	IND	ind	India
15	IDN	idn	Indonesia
16	PHI	phi	Philippines
17	VNM	vnm	Viet Nam
18	UKR	ukr	Ukraine
19	IRN	im	Iran Islamic Republic of
20	UGA	uga	Uganda
21	MDG	mdg	Madagascar
22	RNDC	nzl khm bgd Ika can arg col ecu pry per hnd dom jam tto gbr che nor alb xee kaz kgz tjk aze geo isr jor omn mar tun ben cmr civ gha nga sen tgo xcf xac eth ken mus tza zwe nam	Rest of the regions subject to National Determined Contributions in our simulations
23	ROW	xoc hkg mng twn xea bm lao xse npl pak xsa xna bol chi ury ven xsm cri gtm nic pan slv xca pri xcb xef blr xer xsu arm bhr kwt qat sau tur are xws egy xnf bfa gin xwf mwi moz rwa zmb xec bwa xsc xtw	Rest of the World

**Table 4 ijerph-20-01332-t004:** 2030 NDC greenhouse gas emissions reduction targets by region.

Region	NDC (%)
CHN	−12.62
USA	−45.02
EU	−36.74
JPN	−40.01
KOR	−30.23
AUS	−20.16
SGP	−32.59
BRA	−20.01
MEX	−21.99
MYS	56.60
RUS	16.66
THA	−20.00
ZAF	14.72
IND	36.22
IDN	−29.10
PHI	−2.71
VNM	−8.00
UKR	31.22
IRN	−4.00
UGA	0.00
MDG	−14.00
RNDC	−20.48

Source: calculated based on data from the IMF [55]. Note: The 2030 NDC greenhouse gas emissions reduction targets are relative to the baseline scenario (percentage change).

**Table 5 ijerph-20-01332-t005:** Decarbonisation effects of emissions trading systems in various scenarios.

Linked Regions (%)			China (%)		
Region	Domestic ETSs of Linked Regions	Bilateral ETSs of Linked Regions	Scenario	China’s Domestic ETSs	China’s Bilateral ETSs
USA	−45.02	−18.84	chus	−12.62	−18.84
EU	−36.74	−14.71	cheu	−12.62	−14.71
JPN	−40.01	−13.78	chjp	−12.62	−13.78
KOR	−30.23	−13.12	chkr	−12.62	−13.12
AUS	−20.16	−12.84	chau	−12.62	−12.84
SGP	−32.59	−12.68	chsg	−12.62	−12.68
BRA	−20.01	−12.73	chbr	−12.62	−12.73
MEX	−21.99	−12.87	chmx	−12.62	−12.87
MYS	0.00	−11.41	chmy	−12.62	−11.41
RUS	0.00	−10.55	chru	−12.62	−10.55
THA	−20.00	−12.74	chth	−12.62	−12.74
ZAF	0.00	−11.90	chza	−12.62	−11.90
IND	0.00	−2.91	chin	−12.62	−2.91
IDN	−29.10	−13.45	chid	−12.62	−13.45
PHI	−2.71	−12.53	chpi	−12.62	−12.53
VNM	−8.00	−12.56	chvn	−12.62	−12.56
UKR	0.00	−12.36	chua	−12.62	−12.36
IRN	−4.00	−12.36	chir	−12.62	−12.36
UGA	0.00	−12.62	chug	−12.62	−12.62
MDG	−14.00	−12.62	chmg	−12.62	−12.62

Note: chus, China’s ETS linked with the USA’s; cheu, China’s ETS linked with the EU’s; chjp, China’s ETS linked with Japan’s; chsg, China’s ETS linked with Singapore’s; chkr, China’s ETS linked with Korea’s; chau, China’s ETS linked with Australia’s; chbr, China’s ETS linked with Brazil’s; chru, China’s ETS linked with Russia’s; chth, China’s ETS linked with Thailand’s; chza, China’s ETS linked with South Africa’s; chme, China’s ETS linked with Mexico’s; chmy, China’s ETS linked with Malaysia’s; chin, China’s ETS linked with India’s; chid, China’s ETS linked with Indonesia’s; chvn, China’s ETS linked with Vietnam’s; chpi, China’s ETS linked with the Philippines’; chir, China’s ETS linked with Iran’s; chur, China’s ETS linked with Ukraine’s; chmg, China’s ETS linked with Madagascar’s; chug, China’s ETS linked with Uganda’s; dashed lines represent the equivalent variations in the wrqo scenario.

## Data Availability

Data are not available upon request due to restrictions, e.g., privacy or ethical restrictions.

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
