# Peer review of "Identifying Emissions Reduction Opportunities in International Bilateral Emissions Trading Systems to Achieve China’s Energy Sector NDCs"

_ijerph, 2023, doi:10.3390/ijerph20021332_

Round 1
Reviewer 1 Report
Thank you so much for the opportunity to review this manuscript. Although the research method is not novel but the analysis based different trading scenario is interesting. There are some minor and major concerns that the authors could give consideration based on my reading of the paper, as follows. In addition, the English of the article must be extensively revised. It is too hard to read now.
Abstract and keywords
1. The purpose of the research is not clearly defined in abstract.
2. The research contribution of this paper should be stated in the abstract.
Introduction
3. The introduction part needs more global context.
4. The authors did not explain properly why they took up this topic, what they want to get, what they strive for through their analyze, why this research is important, etc.
5. Although there is the Literature Review section, the knowledge gap is necessary to introduce in this part.
Literature Review
6.This section is suggested to reorganize that closely with the main issue of the article.
Framework
7. Figure 1 is the key to introduce the framework of this article, however, this figure is mess. Maybe paint some color would help.
8. The simulation in Figure 1 lacks a theoretical basis. It is very necessary for the whole article.
Results
9. The note under Figure 2 is suggested to list as a table to express more clearly.
10. The results must be discussed in relation to the literature. Currently, there are too few references in the results and discussion part. Maybe you can separate the discussion from the results.
Conclusion
11. Please write a paragraph about limitations and future research.
Other issues
12.The structure of the paper is loose and lacks logic.
13. English expressions need extensive modification. Such as Line 137-140 is hard to read and long-winded.
14. Recheck the references and their style are according to the journal requirements, and in-text and end-text should be the same and vice versa.
Author Response
Response to Reviewer 1:
Dear reviewer:
Thank you so much for your reviewing of this paper, and I totally accept your suggestion and deeply revised my paper according to your suggestions. And your suggestion indeed highly improves the quality of our paper and make it more readable. Following are my responses to your review reports.
Response:
Thank you so much for the opportunity to review this manuscript. Although the research method is not novel but the analysis based different trading scenario is interesting. There are some minor and major concerns that the authors could give consideration based on my reading of the paper, as follows. In addition, the English of the article must be extensively revised. It is too hard to read now.
Author response:
We appreciate your confirmation of this paper's interest. Regarding language problems, we first thoroughly revise our writing and then use the Language Editing Services of MDPI.
Abstract and keywords
- The purpose of the research is not clearly defined in abstract.
- The research contribution of this paper should be stated in the abstract.
Author response:
This research aims to answer the following questions to find opportunities for China's energy sector to reduce emissions at a lower cost.:
- Does China's energy sector benefit from increased freedom to emit GHGs or lower-cost emission opportunities thanks to the international ETS (Emission trading system) linkage?
- The second question is whether or not linking China's ETS to any region will reduce China's emission costs for its energy sectors if such low-cost emission opportunities exist.
- Third, it has been suggested that linking not all proposed regions to China's ETS results in cheaper emission costs for China's energy sectors. Consequently, what kinds of areas would lower China's emission cost for its energy sectors?
- We then detail why connecting to some regions will lower China's emission cost for energy sectors while connecting to others will not.
- Fifthly, how will ETS and ETS-impact linking's on emission costs influence the Chinese domestic and global energy markets
Does the expansion of green technology, like using electricity to replace fossil fuels, influence the effects of ETS-linking?
[We agree with the reviewer's recommendation and have revised the abstract to explain this work's significance better.]
Introduction
- The introduction part needs more global context.
- The authors did not explain properly why they took up this topic, what they want to get, what they strive for through their analyze, why this research is important, etc.
- Although there is the Literature Review section, the knowledge gap is necessary to introduce in this part.
Author response:
We agree with the reviewer's recommendation that the introduction is revised to explain the article's research topic better and provide some required context.
Literature Review
6.This section is suggested to reorganize that closely with the main issue of the article.
Author response:
We agree with the reviewer's recommendation that the literature is reorganized to be more directly related to the article's main topic.
Framework
- Figure 1 is the key to introduce the framework of this article, however, this figure is mess. Maybe paint some color would help.
- The simulation in Figure 1 lacks a theoretical basis. It is very necessary for the whole article.
Author response:
To improve Figure 1's readability, we have adopted the reviewer's recommendation to add color.
To make the description of the model we employed more coherent, we rewrote section 3.1 and added the Walras rule, the theoretical foundation of the CGE model (depicted in Figure 1). We also provide several crucial functions (equations), and details about those are provided below. Because they would only serve to divert the reader's attention, we do not include them in the document.
Theoretical framework for models: Emission and emission trading settings[key functions used in this model]
According to McDougall and Golub (2009) and Nong (2020), the theoretical foundation for the models that we use in our simulations is as follows:
- Emission accounting
After incorporating non-CO2 greenhouse gas emissions into the GTAP-E database on a region-by-region, commodity-by-commodity, and use-by-use basis, the GTAP-E-Powers model can read GHG emissions from the coefficient. indicates the carbon dioxide emissions from firms' consumption of domestic products; is the noncarbon dioxide GHG emissions from firms' consumption of domestic products; is the carbon dioxide emissions from firms' consumption of imports; is the noncarbon dioxide GHG emissions from firms' consumption of imports; and is the carbon dioxide emissions from private consumption of domestic products.
The corresponding variables have been defined as follows: is the carbon dioxide emissions from firms' consumption of domestic products, is the noncarbon dioxide GHG emissions from firms' consumption of domestic products, and is the carbon dioxide emissions from private consumption of domestic products, and so on. We assume that emissions are proportional to consumption.
Eq.( B.1 )
Eq.( B.2 )
We calculate the carbon dioxide ( ) and noncarbon dioxide ( ) emissions growth for each region and commodity by aggregating the following uses.
Eq.( B.3 )
Eq.( B.4 )
We calculate the total GHG emissions by region, , by adding commodities and the global emissions, , by adding regions.
Eq.( B.5 )
Eq.( B.6 )
- International emissions trade
To represent emission trading, the world is divided into blocs of trading regions; a nontrading region is simply a single-region bloc. Without trade, the set BLOC of blocs is simply a collection of regions; in scenarios involving bilateral ETS linking, China and its emission trading partner would form a single bloc, while the remaining regions would form individual blocs. The REGTOBLOC map depicts the region's division into blocs.
The ETS sectors are divided into groups, with each group representing one of the trading markets covered by the ETS in that block, and these groups are combined to form the CARIND set. For example, if each bloc (e.g., the EU) has two emission trading markets, one for energy-intensive sectors and another for non-energy-intensive sectors, the set of CARIND contains two elements. However, because all sectors covered by ETSs trade carbon permits through a single centralized market (termed CAR_IND) in our simulation, the CARIND set contains only one element (CAR_IND).
While regions and industries' actual GHG emissions, , and GHG emission quotas, , may diverge under emission trading, bloc- and group-level actual emissions, , and emission quotas, , must agree. Because emission trading equalizes the carbon price across blocs and groups, the carbon price, , is a variable at the bloc and group levels.
To enable imposing or relaxing emission constraints, we define the power-of-purchases variable, pempb_e, at the bloc and group level as
Eq.( B.7 )
The emission constraints can then be imposed by making pempb_e exogenous and c_SEC_CTAX endogenous, and they can be relaxed by making endogenous and c_SEC_CTAX exogenous.
When constraints are in place, exogenous quotas at the region and sector levels, , are used, while the endogenous quotas at the bloc and group levels, , are determined by adding up the equations. Without constraints, the quota variables are meaningless, but we must still determine them to solve the model. To accomplish this, we introduce the equation
Eq.( B.8 )
which relates region- and sector-level quotas to actual emissions via a region- and sector-level power-of-purchases variable, pemp_e. The decoupling of regional and sectoral emissions and emission quotas is achieved: in the absence of constraint conditions by making exogenous and endogenous; in the presence of constraints, we make endogenous and exogenous. For each bloc and group, the total quotas are calculated by adding together the quotas for each sector and region.
In order to achieve the decoupling of regional and sectoral emissions and emission quotas, we make endogenous and exogenous in the absence of constraint conditions. In the presence of constraints, we make endogenous and exogenous. For each bloc and group, the total quotas are calculated by adding together the quotas for each sector and region.
- Carbon price
As previously stated, an economic environment devoid of emission constraints can be simulated by endogenizing the emission purchasing power at the bloc and group level and exogenizing the carbon price .
Between market and agent prices in sectors affected by the ETS, there are two wedges: the old ad valorem tax and the new carbon price. To differentiate them, a new valuation level has been established that includes non-carbon prices but excludes carbon prices. The following coefficients have been defined at this level: for firm domestic product consumption, for firm import product consumption, for private domestic product consumption, and so on, by reading them from new arrays in the data file. A modification has been made to the price linking equation to account for the carbon price; for example, the domestic product price for firms is now:
Eq.( B.9 );
where denotes the share of carbon-price-free value to carbon-taxed value, , denotes carbon dioxide emission intensity, , and denotes non-carbon dioxide GHG emission intensity, . Notably, this reduces to the standard GTAP equation for sectors excluded from the ETS, , when both the initial level of carbon tax revenue is zero, implying that is equal to one, and the change in the carbon tax rate is zero.
In the region-level tax revenue variables, for tax on intermediate use, for tax on private consumption, and so forth, carbon tax revenue has been excluded.
- Adjusting regional income to include net revenue from emission trading
The variable denotes the percentage change in the GHG emission quota. The variable, , denotes the change in net revenue from emissions trading for the r region and j sector:
Eq.( B.10 )
is a variable that indicates the change in revenue from net emission trading in the r region:
Eq.( B.11 )
The variable denotes the trade balance, which includes net emission trading revenue:
Eq.( B.12 )
Emission trading has an effect on regional income as well.
Eq.( B.13 )
Due to the absence of carbon tax in the region-wide indirect tax revenue variable, ,it was calculated separately using the linear variable, , and the levels variable, .
The revenue, ,generated by the net emission trading system benefits welfare. This contribution is denoted by a new variable in the model.
- Production
A new production system has been introduced, one that incorporates a greater number of intermediate nesting levels and that combines capital and energy rather than other endowments. To implement this system, a set of subproducts has been defined to correspond to the various composites, including the value-added-energy composite, the capital-energy composite, the energy composite, the non-electricity energy composite, the non-coal energy composite within the non-electricity composite and the electricity composite. Additionally, a set of subproducts for electricity generation has been defined. Subproducts of have been included in a set of commodities demanded by firms alongside endowments and tradables. The variables, , and, , represent the firms' price and demand for tradables, respectively, while and represent the firms' price and demand for . is a variable that represents technological change at each stage of the production system.
For each nest in the production system, a set of inputs and a substitution elasticity has been defined. For non-electricity energy, for example, the set has been defined, comprising the tradable commodity coal and the subproduct (non-coal). The substitution elasticity also been defined, reading its values from a new array EFNL in the parameters file. With these, the demand equation for inputs into non-electricity energy subproduction has been write as:
Eq.( B.14 )
where input is an index into that ranges all elements. The same equation applies to all other nests all other nests in the production system, regardless of whether the inputs are tradable, endowments, subproducts, or some combination thereof.
Specifically, the set has been defined to encompass seven different types of base load electricity: NuclearBL, CoalBL, GasBL, HydroBL, OilBL, WindBL, and OtherBL; and has been defined to encompass four different types of peak load electricity: GasP, HydroP, OilP, and SolarP.
Results
- The note under Figure 2 is suggested to list as a table to express more clearly.
- The results must be discussed in relation to the literature. Currently, there are too few references in the results and discussion part. Maybe you can separate the discussion from the results.
Author response:
Following the reviewer's recommendation, we have swapped out Figure 2 for a table that will be more informative.
More than that, we compare and contrast the results of the publications by referencing relevant literature.
Conclusion
- Please write a paragraph about limitations and future research.
Author response:
We finally take the reviewer's advice and include a section on our limitations and possibilities for future study.
Other issues
12.The structure of the paper is loose and lacks logic.
- English expressions need extensive modification. Such as Line 137-140 is hard to read and long-winded.
- Recheck the references and their style are according to the journal requirements, and in-text and end-text should be the same and vice versa.
Author response:
To help this paper flow better, we rewrote and reorganized some of the material mentioned above. We restructure this text using fewer words to show how we mean to express ourselves and to maintain a high level of English. Attached is MDPI's certification form confirming that we used their Language Editing Services. We double-check the citations and formatting to ensure they adhere to the journal's standards.

Reviewer 2 Report
Highlight changes in yellow in a next revision, please. No track changes.
I would suggest revising the language because I believe this cannot be said like this.
“This study investigates these prospects by simulating the 13
effect”
place address or abbreviations at first use, both abstract as manuscript.
“GTAP”
I would suggest revising this language referring to findings instead of conclusions and do not “LIST”.
“This re- 16
search contributes to the following conclusions. (1)”
it would be important to end the abstract in a significant way to understand why is this important. Particularly in the context of the journal that the authors have chosen to publish in. Until now, I see no real connection between the journal and the subject being addressed because the sustainability or the environmental aspects are not being. Emphasized. As well as health, perhaps.
Please make sure to correct all these references because it cannot appear after the dot.
“inflation.[2, 3] ”
as I always say, this is not a thesis, so I see no real interest in presenting this content like this.
“This article's remaining content is structured as follows. Section 2”
instead, please clarify the objectives of this study.
I would ask authors not to use the WE personal formal of address.
“We examine”
when using this term robust. Authors need to clarify why is it being used supported in relevant literature.
“Robust testing examines ”
please do not start by lower letter and please do not use the word THE inside headings.
Are these already results? Then please be very careful. With what content to include within each section.
“In our simulation, as showed in figure 1, the total ETS emissions and the total ETS 195
emission quota, both of which can be modified via ETS linkage, determine the decarbon- 196
ization duties with of ETS and, by extension, the carbon price endogenized in the ETS.”
I would like to see many more references, in particular parts of the text where authors are clarifying content.
“The income effect of the carbon price 212
on energy demand indicates that the gross regional income, which is affected by the in- 213
crease in the carbon price, will decrease energy demand and, consequently, energy de- 214
mand price. The first three mechanisms can be grouped into the cost effect, which influ- 215
ence the overall cost of energy usage brought about by the shift in carbon price influenced 216
by ETS and ETS linking. Market clearing of energy prices is always endogenized by the 217
three cost effects combined with the income effect.”
As in headings , please do not use the term THE and make sure that every figures are original.
So please see the term being used in either plural or singular.
“Thus, our models show”
please check the spacing all over the entire manuscript.
“Indonesia(IDN),”
authors need to limit the necessary information in the caption. There is no need to mention the percentage which can be included in the headings inside the table.
“
Table 1: NDC submitters' 2030 greenhouse gas emission reduction targets relative to |
271 |
”
of course the results are being analyzed, so just limit the heading to results and discussion, not discussion and results.
“4. Analysis and discussion of the results”
again notice is here. So try to use meaningful language.
“In the following subsections, we will examine in greater detail”
A section or subsection cannot start with figures it it needs to have a text linking it all.
“4.1The influence of the implementation and linking of ETS on the carbon price 344
Figure 2 Emissions Trading System's Decarbonization Duties in Various Scenarios”
In all cases where you have more than one figure, each one needs to be identified by a different letter and a sub caption must correspond to each letter after the main caption.
See that there is no need to include the word different inside the caption as well as results, because the figure is included in the results section. So authors should see some papers to be able to understand how at international level they use. Of a proper language is important in scientific publications.
“Figure 3 Different ETSs resul”
Again, this is a grouped figure.
Again, this is also a grouped figure. Also be careful to include the two from CO2 in subscript.
Also inside the figure.
“Figure 5 Sectoral emissions and carbon emission permit trading at the regional level”
and again it cannot start with a figure.
“4.2China’s domestical energy market”
values presented like this are not really readable, particularly because authors have not even respect the proper spacing.
“tricity transfer and distribution, will increase 551
34.04%,7.58%,5.65%,3.09%,3.26%,1.87%,1.76%,1.62%,1.60%,1.60%,1.41%.”
an arrow could also be used for example.
Again, check proper spacing.
“4.3China’s international energy trade”
and again the use of the term robust is not clarifying. Also, the reader should be able to look at the heading and find it self explanatory.
“4.5Robust test: electricity substitute”
some authors opt to present a huge number of figures. It should be questioned whether they are all necessary and relevant because as at some time they distract the reader from the important points being pointed out through the paper.
I would ask authors to consider that the conclusions section is very important. It needs to contextualize the article being presented so that the reader understands why it is important, then a brief methodology, then main findings and practical implications, limitations and future prospects.
Please do not start a new paragraph with AND…
“And the "
in a scientific article. The conclusions should necessarily be briefer so that the reader may focus in what it is important. I would suggest part of this content to be moved to other sections of the article.
Until now, I believe that authors were not able to successfully link the subject being studied. In terms of environmental implications with the journal itself.
The term sustainability, for example, is never used.
The term environment is not used in a direct way.
I would also like to see some important values expressed quantitatively in the conclusions.
Authors should consider to choose how supplementary material is being presented, not inside the text as appendix, but in another part and what is absolutely necessary to be considered. Also consider the comments from tables made above to the tables appearing in the appendix. You cannot call it, for example, Table 2. Because it is a table that does not make part of the main text.
Consider that the references are extremely important in terms of an international index journal, so I would ask you to include more references from more international author as well as recent references to. I’m not referring to sites.
I hope the authors are able to understand that despite the important work developed and presented, they need to link what is being presented to the specific scope of this journal, and that the comments are intended to assist the authors in achieving that.
There is some similarity found in the text that it is not followed by the necessary references.
I can also see that in a specific tables of the appendix there are content that can be found in other publications and no reference at all is used to accompany the captions.
Author Response
Response to Reviewer 2:
Dear reviewer:
Thank you so much for your reviewing of this paper, and I totally accept your suggestion and deeply revised my paper according to your suggestions. And your suggestion indeed highly improves the quality of our paper and make it more readable. Following are my responses to your review reports.
Response:
Highlight changes in yellow in a next revision, please. No track changes.
Author response:
In response to the IJERPH Editorial Office's request that I submit my revised manuscript with "Track Changes" enabled, I am attaching two versions of the same document: one attached below with my revisions highlighted in yellow, for you to review and one with their revisions using “Track Changes" enabled for the IJERPH Editorial Office., for the IJERPH Editorial Office.
I would suggest revising the language because I believe this cannot be said like this.
“This study investigates these prospects by simulating the 13
effect”
Author response:
After a thorough round of revisions to address the paper's linguistic issues, we enlist the help of MDPI's Language Editing Services to finalize the paper's polished tone.
place address or abbreviations at first use, both abstract as manuscript.
“GTAP”
Author response:
We add the abbreviation's full form in brackets following the first occurrence of the abbreviation in this document.
I would suggest revising this language referring to findings instead of conclusions and do not “LIST”.
“This re- 16
search contributes to the following conclusions. (1)”
Author response:
We have changed the word entirely in light of the reviewer's comments.
it would be important to end the abstract in a significant way to understand why is this important. Particularly in the context of the journal that the authors have chosen to publish in. Until now, I see no real connection between the journal and the subject being addressed because the sustainability or the environmental aspects are not being. Emphasized. As well as health, perhaps.
Author response:
We take on board all of the reviewer's comments and suggestions, and we stress once more the significance of our article to environmental protection and sustainable development in China.
Please make sure to correct all these references because it cannot appear after the dot.
“inflation.[2, 3] ”
Author response:
Following the reviewer's advice, we updated all of the citations in the text.
as I always say, this is not a thesis, so I see no real interest in presenting this content like this.
“This article's remaining content is structured as follows. Section 2”
instead, please clarify the objectives of this study.
Author response:
We agree with the reviewer's suggestion to remove this meaningless standard from the dissertation's introduction and replace it with a section outlining the research goals.
I would ask authors not to use the WE personal formal of address.
“We examine”
Author response:
We agree with the reviewer's suggestion to replace subjective pronouns like "we" with more objective language such as "this paper," "this study," and "experiments conducted in this study."
when using this term robust. Authors need to clarify why is it being used supported in relevant literature.
“Robust testing examines”
Author response:
To show that robust testing (also called sensitive analysis) is necessary, we reorganize our paper and include new references.
please do not start by lower letter and please do not use the word THE inside headings.
Author response:
We fully adopt the reviewer’s suggestions.
Are these already results? Then please be very careful. With what content to include within each section.
“In our simulation, as showed in figure 1, the total ETS emissions and the total ETS 195
emission quota, both of which can be modified via ETS linkage, determine the decarbon- 196
ization duties with of ETS and, by extension, the carbon price endogenized in the ETS.”
Author response:
Sincerely, we apologise if this has been an inconvenient read for you. These methods are the backbone of experimental and simulated research into economic behaviour. [We revise this section to provide a more straightforward explanation.]
I would like to see many more references, in particular parts of the text where authors are clarifying content.
“The income effect of the carbon price 212
on energy demand indicates that the gross regional income, which is affected by the in- 213
crease in the carbon price, will decrease energy demand and, consequently, energy de- 214
mand price. The first three mechanisms can be grouped into the cost effect, which influ- 215
ence the overall cost of energy usage brought about by the shift in carbon price influenced 216
by ETS and ETS linking. Market clearing of energy prices is always endogenized by the 217
three cost effects combined with the income effect.”
Author response:
In fact, this section is a contribution of this paper that has not been adequately addressed in other papers, and we have rewrite this part to make it more clearly. [This section is revised to include references to income effect.].
As in headings, please do not use the term THE and make sure that every figures are original.
Author response:
We adopt the reviewer's recommendations in full, and we are confident that all figures are original. [We check headings to ensure that THE is not used.].
So please see the term being used in either plural or singular.
“Thus, our models show”
Author response:
We check the grammar of single and plural in this work and replace "models" with "simulations" for greater precision.
please check the spacing all over the entire manuscript.
“Indonesia(IDN),”
Author response:
We adopt all of the reviewer's recommendations and rewrite and check the spacing across the entire text.
authors need to limit the necessary information in the caption. There is no need to mention the percentage which can be included in the headings inside the table.
“
Table 1: NDC submitters' 2030 greenhouse gas emission reduction targets relative to |
271 |
”
Author response:
We condense the caption to highlight only the essential information and provide an explanation in the note.
of course the results are being analyzed, so just limit the heading to results and discussion, not discussion and results.
“4. Analysis and discussion of the results”
We rewrite the title as suggested by the reviewers.
again notice is here. So try to use meaningful language.
“In the following subsections, we will examine in greater detail”
Author response:
This section is rewritten using meaningful language.
A section or subsection cannot start with figures it it needs to have a text linking it all.
“4.1The influence of the implementation and linking of ETS on the carbon price 344
Figure 2 Emissions Trading System's Decarbonization Duties in Various Scenarios”
In all cases where you have more than one figure, each one needs to be identified by a different letter and a sub caption must correspond to each letter after the main caption.
Author response:
We agree with the reviewer's suggestion to avoid starting sections or subsections with figures by linking them with text.
We implement the reviewer's suggestion to identify each subfigure with a Roman numeral sub-number by appending the sub-number to the grouped figure.
See that there is no need to include the word different inside the caption as well as results, because the figure is included in the results section. So authors should see some papers to be able to understand how at international level they use. Of a proper language is important in scientific publications.
“Figure 3 Different ETSs resul”
Again, this is a grouped figure.
Author response:
We rewrite the Figure caption.
We implement the reviewer's suggestion to identify each subfigure with a Roman numeral sub-number by appending the sub-number to the grouped figure.
Again, this is also a grouped figure. Also be careful to include the two from CO2 in subscript.
Also inside the figure.
“Figure 5 Sectoral emissions and carbon emission permit trading at the regional level”
Author response:
We modify 2 to be the subscript of CO2.
and again it cannot start with a figure.
“4.2China’s domestical energy market”
Author response:
We accept the reviewer's suggestion to avoid starting sections or subsections with figures by linking them with text.
values presented like this are not really readable, particularly because authors have not even respect the proper spacing.
“tricity transfer and distribution, will increase 551
34.04%,7.58%,5.65%,3.09%,3.26%,1.87%,1.76%,1.62%,1.60%,1.60%,1.41%.”
an arrow could also be used for example.
Author response:
We adopt the reviewer's suggestions and revise and check the spacing throughout the entire manuscript.
Again, check proper spacing.
“4.3China’s international energy trade”
Author response:
We adopt the reviewer's suggestions and revise and check the spacing throughout the entire manuscript.
and again the use of the term robust is not clarifying. Also, the reader should be able to look at the heading and find it self explanatory.
“4.5Robust test: electricity substitute”
Author response:
The first sentence of this section has been revised to provide context regarding the need for the "robust" test.
some authors opt to present a huge number of figures. It should be questioned whether they are all necessary and relevant because as at some time they distract the reader from the important points being pointed out through the paper.
Author response:
In light of the reviewer's comments that including specific figures in the paper would cause the audience to lose focus on the main arguments, we decided to remove the last group of figures.
I would ask authors to consider that the conclusions section is very important. It needs to contextualize the article being presented so that the reader understands why it is important, then a brief methodology, then main findings and practical implications, limitations and future prospects.
Author response:
In response to the reviewer's recommendation, we rewrote a portion of the conclusion to emphasize our findings better. We also include a paragraph that discusses our publications' limitation and potential future developments.
Please do not start a new paragraph with AND…
“And the "
in a scientific article. The conclusions should necessarily be briefer so that the reader may focus in what it is important. I would suggest part of this content to be moved to other sections of the article.
Author response:
Following the reviewer's advice, we have rewritten our conclusion to be shorter and more succinct, as is appropriate given the nature of the work.
Until now, I believe that authors were not able to successfully link the subject being studied. In terms of environmental implications with the journal itself.
The term sustainability, for example, is never used.
The term environment is not used in a direct way.
I would also like to see some important values expressed quantitatively in the conclusions.
Author response:
We revised the paper to better fit the journal's mission by adding quantitative data on China's decarbonization(environment) to the discussion and conclusions and by stressing the paper's contribution to environmental protection and sustainable development.
Authors should consider to choose how supplementary material is being presented, not inside the text as appendix, but in another part and what is absolutely necessary to be considered. Also consider the comments from tables made above to the tables appearing in the appendix. You cannot call it, for example, Table 2. Because it is a table that does not make part of the main text.
Author response:
Keeping these tables in the appendix was a conscious decision because they will be helpful to the reader in grasping the scenario design. Tables in the appendix have been renumbered using Roman numerals per reviewers' suggestions. We emphasized the paper's contribution to environmental protection and sustainable development in the revised manuscript to fit the journal's aims and objectives better.
Consider that the references are extremely important in terms of an international index journal, so I would ask you to include more references from more international author as well as recent references to. I’m not referring to sites.
I hope the authors are able to understand that despite the important work developed and presented, they need to link what is being presented to the specific scope of this journal, and that the comments are intended to assist the authors in achieving that.
Author response:
We have added new references to improve the paper's overall depth of analysis.
In addition, we revised the manuscript to better fit the journal's aims by emphasizing the paper's contribution to environmental protection and sustainable development.
There is some similarity found in the text that it is not followed by the necessary references.
Author response:
The new references we have added to help the paper's overall clarity.
I can also see that in a specific table of the appendix there are content that can be found in other publications and no reference at all is used to accompany the captions.
Author response:
These tables are specific to this research and may look familiar but differ from those in other works. Nonetheless, per the reviewer's suggestion, we have included some references that went into making these tables.

Reviewer 3 Report
The work needs a significant language review as the beauty and message are not showing because of the current style and language, There is a need for more relevant literature, a clear methodology etc... See attachment.

Author Response
Dear reviewer:
Thanks again for taking the time to review this paper; I appreciate the feedback very much and have made extensive changes in light of your comments. Moreover, the quality of our paper and its readability are both greatly enhanced by your suggestion. My comments on your review reports are below.
Response:
The work needs a significant language review as the beauty and message are not showing because of the current style and language, there is a need for more relevant literature, a clear methodology etc... See attachment. There is a need for major grammar check. the current flow makes it difficult to follow the argument you are trying to bring to fore. Also try and follow the conventional abstract presentation of Introduction/background, methodology, findings and recommendation.
Author response:
To address the reviewer's concerns about the paper's language, we have used the Language Editing Services of MDPI.
We also rewrite the paper extensively to include more context information, a more transparent demonstration of methodology, and a more logical presentation of its findings.

Reviewer 4 Report
The article is written on a relevant topic, well structured, has an adequate bibliography.
However, there are a number of significant remarks, the elimination of which will improve the article quality :
1. The authors propose a GTAP-E-Powers model, but nowhere are the methods, tools that are used in the context of this model. Based on this, it is not clear how the results proposed by the authors can be evaluated. How do the results obtained by the authors differ from the known ones.
2. The set of designations on the mechanism, Figure 1, are not clear, they are not deciphered anywhere. This makes scheme 1 not informative. What is the difference between entities in squares and entities in ovals in the mechanism in Figure 1?
3. How can the results obtained in this work be used to make decisions on the management of decarbonization from the standpoint of business and the govenment ?
Based on the foregoing, I believe that the article should be revised and substantially supplemented.
Author Response
Dear reviewer:
Thank you so much for your reviewing of this paper, and I totally accept your suggestion and deeply revised my paper according to your suggestions. And your suggestion indeed highly improves the quality of our paper and make it more readable. Following are my responses to your review reports.
Response:
The article is written on a relevant topic, well structured, has an adequate bibliography.
Author response:
We appreciate your confirmation of this paper's work.
However, there are a number of significant remarks, the elimination of which will improve the article quality :
- The authors propose a GTAP-E-Powers model, but nowhere are the methods, tools that are used in the context of this model. Based on this, it is not clear how the results proposed by the authors can be evaluated. How do the results obtained by the authors differ from the known ones.
Author response:
GTAP-E-Powers is a Linear Programming Model and a general equilibrium model based on Microeconomy and Macroeconomy theory (https://www.gov.scot/publications/cge-modelling-introduction) that simulates the impact of policy on the Macroeconomy. Multiple programming languages and environments, including R, Python, MATLAB, gams, and GEMPACK, can execute these simulations. We relied on GEMPACK, a programme developed by the COPS, to run the simulation. Our contribution is applying an existing model to the solution of a novel problem, as demonstrated in the concluding section of the relevant literature review.
[To make things clearer, we reorganised the literature review and Section 3.1, which was intended to introduce the methods and tools used in this model. In addition, the essential functions of the model are listed below.]
Theoretical framework for models: Emission and emission trading settings[key functions used in this model]
According to McDougall and Golub (2009) and Nong (2020), the theoretical foundation for the models that we use in our simulations is as follows:
- Emission accounting
After incorporating non-CO2 greenhouse gas emissions into the GTAP-E database on a region-by-region, commodity-by-commodity, and use-by-use basis, the GTAP-E-Powers model can read GHG emissions from the coefficient. indicates the carbon dioxide emissions from firms' consumption of domestic products; is the noncarbon dioxide GHG emissions from firms' consumption of domestic products; is the carbon dioxide emissions from firms' consumption of imports; is the noncarbon dioxide GHG emissions from firms' consumption of imports; and is the carbon dioxide emissions from private consumption of domestic products.
The corresponding variables have been defined as follows: is the carbon dioxide emissions from firms' consumption of domestic products, is the noncarbon dioxide GHG emissions from firms' consumption of domestic products, and is the carbon dioxide emissions from private consumption of domestic products, and so on. We assume that emissions are proportional to consumption.
Eq.( B.1 )
Eq.( B.2 )
We calculate the carbon dioxide ( ) and noncarbon dioxide ( ) emissions growth for each region and commodity by aggregating the following uses.
Eq.( B.3 )
Eq.( B.4 )
We calculate the total GHG emissions by region, , by adding commodities and the global emissions, , by adding regions.
Eq.( B.5 )
Eq.( B.6 )
- International emissions trade
To represent emission trading, the world is divided into blocs of trading regions; a nontrading region is simply a single-region bloc. Without trade, the set BLOC of blocs is simply a collection of regions; in scenarios involving bilateral ETS linking, China and its emission trading partner would form a single bloc, while the remaining regions would form individual blocs. The REGTOBLOC map depicts the region's division into blocs.
The ETS sectors are divided into groups, with each group representing one of the trading markets covered by the ETS in that block, and these groups are combined to form the CARIND set. For example, if each bloc (e.g., the EU) has two emission trading markets, one for energy-intensive sectors and another for non-energy-intensive sectors, the set of CARIND contains two elements. However, because all sectors covered by ETSs trade carbon permits through a single centralized market (termed CAR_IND) in our simulation, the CARIND set contains only one element (CAR_IND).
While regions and industries' actual GHG emissions, , and GHG emission quotas, , may diverge under emission trading, bloc- and group-level actual emissions, , and emission quotas, , must agree. Because emission trading equalizes the carbon price across blocs and groups, the carbon price, , is a variable at the bloc and group levels.
To enable imposing or relaxing emission constraints, we define the power-of-purchases variable, pempb_e, at the bloc and group level as
Eq.( B.7 )
The emission constraints can then be imposed by making pempb_e exogenous and c_SEC_CTAX endogenous, and they can be relaxed by making endogenous and c_SEC_CTAX exogenous.
When constraints are in place, exogenous quotas at the region and sector levels, , are used, while the endogenous quotas at the bloc and group levels, , are determined by adding up the equations. Without constraints, the quota variables are meaningless, but we must still determine them to solve the model. To accomplish this, we introduce the equation
Eq.( B.8 )
which relates region- and sector-level quotas to actual emissions via a region- and sector-level power-of-purchases variable, pemp_e. The decoupling of regional and sectoral emissions and emission quotas is achieved: in the absence of constraint conditions by making exogenous and endogenous; in the presence of constraints, we make endogenous and exogenous. For each bloc and group, the total quotas are calculated by adding together the quotas for each sector and region.
In order to achieve the decoupling of regional and sectoral emissions and emission quotas, we make endogenous and exogenous in the absence of constraint conditions. In the presence of constraints, we make endogenous and exogenous. For each bloc and group, the total quotas are calculated by adding together the quotas for each sector and region.
- Carbon price
As previously stated, an economic environment devoid of emission constraints can be simulated by endogenizing the emission purchasing power at the bloc and group level and exogenizing the carbon price .
Between market and agent prices in sectors affected by the ETS, there are two wedges: the old ad valorem tax and the new carbon price. To differentiate them, a new valuation level has been established that includes non-carbon prices but excludes carbon prices. The following coefficients have been defined at this level: for firm domestic product consumption, for firm import product consumption, for private domestic product consumption, and so on, by reading them from new arrays in the data file. A modification has been made to the price linking equation to account for the carbon price; for example, the domestic product price for firms is now:
Eq.( B.9 );
where denotes the share of carbon-price-free value to carbon-taxed value, , denotes carbon dioxide emission intensity, , and denotes non-carbon dioxide GHG emission intensity, . Notably, this reduces to the standard GTAP equation for sectors excluded from the ETS, , when both the initial level of carbon tax revenue is zero, implying that is equal to one, and the change in the carbon tax rate is zero.
In the region-level tax revenue variables, for tax on intermediate use, for tax on private consumption, and so forth, carbon tax revenue has been excluded.
- Adjusting regional income to include net revenue from emission trading
The variable denotes the percentage change in the GHG emission quota. The variable, , denotes the change in net revenue from emissions trading for the r region and j sector:
Eq.( B.10 )
is a variable that indicates the change in revenue from net emission trading in the r region:
Eq.( B.11 )
The variable denotes the trade balance, which includes net emission trading revenue:
Eq.( B.12 )
Emission trading has an effect on regional income as well.
Eq.( B.13 )
Due to the absence of carbon tax in the region-wide indirect tax revenue variable, ,it was calculated separately using the linear variable, , and the levels variable, .
The revenue, ,generated by the net emission trading system benefits welfare. This contribution is denoted by a new variable in the model.
- Production
A new production system has been introduced, one that incorporates a greater number of intermediate nesting levels and that combines capital and energy rather than other endowments. To implement this system, a set of subproducts has been defined to correspond to the various composites, including the value-added-energy composite, the capital-energy composite, the energy composite, the non-electricity energy composite, the non-coal energy composite within the non-electricity composite and the electricity composite. Additionally, a set of subproducts for electricity generation has been defined. Subproducts of have been included in a set of commodities demanded by firms alongside endowments and tradables. The variables, , and, , represent the firms' price and demand for tradables, respectively, while and represent the firms' price and demand for . is a variable that represents technological change at each stage of the production system.
For each nest in the production system, a set of inputs and a substitution elasticity has been defined. For non-electricity energy, for example, the set has been defined, comprising the tradable commodity coal and the subproduct (non-coal). The substitution elasticity also been defined, reading its values from a new array EFNL in the parameters file. With these, the demand equation for inputs into non-electricity energy subproduction has been write as:
Eq.( B.14 )
where input is an index into that ranges all elements. The same equation applies to all other nests all other nests in the production system, regardless of whether the inputs are tradable, endowments, subproducts, or some combination thereof.
Specifically, the set has been defined to encompass seven different types of base load electricity: NuclearBL, CoalBL, GasBL, HydroBL, OilBL, WindBL, and OtherBL; and has been defined to encompass four different types of peak load electricity: GasP, HydroP, OilP, and SolarP.
- The set of designations on the mechanism, Figure 1, are not clear, they are not deciphered anywhere. This makes scheme 1 not informative. What is the difference between entities in squares and entities in ovals in the mechanism in Figure 1?
Author response:
The squares represent the production block of the model, whereas the entities represent the emission trading block.
We rewrite this section to clarify how the linkage of the emission trading system will affect the production sector and the macroeconomy.
3.How can the results obtained in this work be used to make decisions on the management of decarbonization from the standpoint of business and the govenment ?
Author response:
Since September of last year, decarbonization has caused an energy crisis in some areas of China, including a lack of electricity in many cities in Northeast China. Currently, several coal-fired power plants in China are relatively new. The early retirement of such investments before their final depreciation could result in substantial sunk costs for China, which is unacceptable to the investor. This paper seeks low-cost emission opportunities for China's energy sector via international emission trading and the fulfillment of China's international decarbonization commitment. We can reduce China's carbon footprint if we establish a bilateral emission trading system with regions with relatively lower carbon prices than China. It will provide China's energy sector with greater decarbonization flexibility.
[We rework the paper's introductory and concluding sections to highlight better the paper's significance to policymakers and business leaders.]
Based on the foregoing, I believe that the article should be revised and substantially supplemented.
Author response:
This document is revised to make it more readable and meaningful.

Round 2
Reviewer 1 Report
The author has carefully revised the paper, and I think it can be published.
Author Response
Dear reviewer:
We are grateful for your time and effort in reviewing this paper. Your affirmation of its publication would prove my success in this study. More importantly, I am confident that its timely publication will also serve as a valuable suggestion for the Chinese to lower their carbon footprint and foster sustainable development, which is critical for China and the entire world.
Reviewer 2 Report
Highlight changes in yellow in a next revision, please. No track changes.
It would be important to clarify this...
Too general.
“Among these, links to certain regions, such as India 31
and Russia, could lower China's decarbonisation costs to a high degree. In contrast, links to other 32
regions, such as the USA and the EU, could increase China's decarbonisation costs.”
Again, please remove “The” from captions...
“Table 2. The decarbonisation effects of emissions trading systems in various scenarios.”
It seems authors have not understood my comments. grouped figures need to be identified by a different letter and a subcaption must be presented after main caption, by figure letter
“Figure 2 Different ETSs result in . The fluctuating regional prices for carbon permits in various 622
scenarios.”
and several others...
Again, please remove “THE” from every graphic...
Do not repeat the title and the same caption, do not duplicate content...
“Figure 5. The fluctuations in domestic market prices and energy sales in China.”
unless considered crucial, references should be removed from the conclusions section...
“o investors [15, 16].”
I would expect some quantitative data in the abstract and conclusions, due to the data expressed in the manuscript
Once again, the data presented in appendix should be self/explanatory in terms of headings and content
Check spacing all over, again...
“nalysis[48, 49]”
Author Response
Dear reviewer:
We are grateful for your time and effort in reviewing this paper. We are confident that its timely publication will also serve as a valuable suggestion for the Chinese to lower their carbon footprint and foster sustainable development, which is critical for China and the entire world. We have sincerely considered your suggestion and have revised this manuscript one by one. Detailed information to respond reviewer's suggestion is attached as a PDF.

Reviewer 3 Report
The required corrections have been made as advised in the previous review and the current work is publishable.
Author Response

(The authors gave the same response as above.)

Reviewer 4 Report
Significant changes have been made to the article in accodance with rhe reviewer conments.
I believe that the manuscript has been sufficiently improved to warrant publication in the journal.
Author Response

(The authors gave the same response as above.)
